# Noise Separation guided Candidate Label Reconstruction for Noisy Partial Label Learning

**Xiaorui Peng**[1], **Yuheng Jia**[2,3,*] **Fuchao Yang**[1], **Ran Wang**[4,5], **Min-Ling Zhang**[2,6]
[1]College of Software Engineering, Southeast University
[2]School of Computer Science and Engineering, Southeast University
[3]Key Laboratory of New Generation Artificial Intelligence Technology and
Its Interdisciplinary Applications (Southeast University)
[4]Shenzhen Key Laboratory of Advanced Machine Learning and Applications,
School of Mathematical Sciences, Shenzhen University
[5]Guangdong Provincial Key Laboratory of Intelligent Information Processing, Shenzhen University
[6]Key Laboratory of Computer Network and Information Integration (Southeast University)
{xiaorui, yhjia, yangfc, zhangml}@seu.edu.cn; wangran@szu.edu.cn

## Abstract

Partial label learning is a weakly supervised learning problem in which an instance is annotated with a set of candidate labels, among which only one is the correct label. However, in practice the correct label is not always in the candidate label set, leading to the noisy partial label learning (NPLL) problem. In this paper, we theoretically prove that the generalization error of the classifier constructed under NPLL paradigm is bounded by the noise rate and the average length of the candidate label set. Motivated by the theoretical guide, we propose a novel NPLL framework that can separate the noisy samples from the normal samples to reduce the noise rate and reconstruct the shorter candidate label sets for both of them. Extensive experiments on multiple benchmark datasets confirm the efficacy of the proposed method in addressing NPLL. For example, on CIFAR100 dataset with severe noise, our method improves the classification accuracy of the state-of-the-art one by 11.57%. The code is available at: https://github.com/pruirui/PLRC.

## 1 Introduction

Partial label learning (PLL) (Cour et al., 2011; Nguyen & Caruana, 2008; Jiang et al., 2024; Yang et al., 2024) is a weakly supervised learning paradigm (Zhou, 2018) that allows samples to be associated with a set of candidate labels, of which only one is the ground-truth label. Due to the low annotation cost of collecting partial label dataset in real-world scenarios, PLL has attracted significant attention from the community with many applications, such as automatic face annotation (Chen et al., 2018), web mining (Luo & Orabona, 2010), and face age estimation (Panis & Lanitis, 2014). A variety of methods have been developed to tackle the PLL problem, including average-based methods (Hüllermeier & Beringer, 2006; Cour et al., 2009; Jin & Ghahramani, 2002), graph-based methods (Wang et al., 2022a; Jia et al., 2023b; Xu et al., 2019; Zhang et al., 2016) and pseudo-labeling-based methods (Lv et al., 2020; Wen et al., 2021; Jia et al., 2023a; 2024), etc.

Despite the promise, these PLL methods have been driven by the assumption that the ground-truth label must lay in the candidate label set (CLS), which may not hold in practice due to the existence of the non-expert annotators (Li et al., 2020; Hossain & Kauranen, 2015; Shi et al., 2023). Recently, some researchers have turned their attention to a more practical setting called noisy partial label learning (NPLL) (Lv et al., 2024). In NPLL, the ground-truth label of some instances may not lie in the CLS. For ease of description, in this article, samples whose ground-truth label is not included in the CLS are referred as **noisy samples**, and those that do include the ground-truth label are termed as **normal samples**. To address the NPLL task, APLLS (Lv et al., 2024) proposes a family of robust

---

*Corresponding author.

average losses to against the existence of label noise. PiCO+ (Wang et al., 2024) and UPLLRS (Shi et al., 2023) detect noisy samples and treat them as unlabeled samples and exploit them by a semi-supervised learning approach. ALIM (Xu et al., 2023) incorporates non-candidate labels into candidate labels through dynamic weighting to combat the noise problem. Although these methods have achieved commendable results, they have not theoretically elucidated the essence of the NPLL problem.

To this end, we first theoretically construct the generalization error bound of NPLL, where we find that two key issues may help solve the NPLL problem: i) a lower noise rate and ii) a smaller candidate label set. The noise rate is defined as the proportion of noisy samples to the overall samples. In NPLL, the key to reducing the noise rate is correctly detecting noisy samples from the NPLL dataset, because noisy samples can be easily converted into normal ones by exchanging the CLS with non-CLS (ground-truth label must locate at non-CLS for noisy samples). To achieve this goal, we propose a metric called ECK to **distinguish normal samples and noisy samples**. Considering that the model's output in the early stages of training is not reliable for hard samples, we employ the ECK metric to partition samples into three parts: highly reliable normal samples, highly reliable noisy samples, and uncertain samples. As the performance of model improves, we gradually decrease the proportion of uncertain samples to achieve a clearer separation between normal samples and noisy samples. Apart from reducing the noise rate, it is also necessary to decrease the size of the CLS. To achieve this target, we propose to **reconstruct the CLS for each instance**. The restructured CLS aims to achieve two objectives: reducing its size and containing ground-truth label. We propose an instance-adaptive parameter to balance these two objectives, thereby reconstructing a faithful shorter CLS for each instance. Through sample separation and CLS reconstruction, we can reduce the generalization bound, and accordingly promote the classification performance. Our method serves as a plug-in for PLL methods to enhance their performance on NPLL datasets. We extensively evaluate our method on several benchmarks, which validates the efficacy of our method. Our contributions can be summarized as follows:

- For the first time, we provide the generalization error bound of the classifier constructed under NPLL, where we find that lower noise rate and smaller candidate label are two key factors to solve the NPLL problem.

- With insights drawn from the theoretical findings, we propose a novel NPLL method which includes two components: progressive sample separation and CLS reconstruction. Based on those two novel components, our method can effectively reduce the noise rate while simultaneously decreasing the size of CLS.

- Our method can serve as a plug-in for existing PLL methods to enhance their performance on NPLL datasets. Extensive experimental results validate that our method outperforms the current state-of-the-art (SOTA) methods by a large margin, e.g., a **11.57%** improvement on CIFAR100 with extreme noise and ambiguity level.

## 2  PROBLEM SETUP

Let $\mathcal{X} = \mathbb{R}^d$ be the input space of $d$ dimensions and $\mathcal{Y} = \{1, \cdots, C\}$ be the label space, and $C$ be the number of classes. We denote $\mathcal{D} = \{(\boldsymbol{x_i}, Y_i)\}_{i=1}^n$ the training dataset with $n$ samples, and each tuple in $\mathcal{D}$ comprises of a vector $\boldsymbol{x_i} \in \mathcal{X}$ and a CLS $Y_i \subset \mathcal{Y}$. Let $\overline{Y}_i = \mathcal{Y} - Y_i$ represents the non-CLS of $\boldsymbol{x_i}$. In traditional PLL, the ground-truth label $y_i$ is concealed in CLS i.e. $y_i \in Y_i$ (Wang et al., 2022b; Chen et al., 2018), while NPLL allows some samples of which the ground-truth label $y_i$ locates outside the candidate set $Y_i$, that is $y_i \in \overline{Y}_i$ (Lv et al., 2024; Xu et al., 2023). Let $f(\boldsymbol{x}; \theta)$ denote a deep neural network parameterized by $\theta$, which transforms $\boldsymbol{x}$ to into a probability prediction vector $\boldsymbol{p}$. Our goal is to train a multi-class classifier $f(\boldsymbol{x}; \theta)$ using the NPLL dataset $\mathcal{D}$.

## 3  THEORETICAL ANALYSIS

Here, we provide a generalization error bound for NPLL problem to analyze the factors that can enhance the generalization ability of the model. The true risk with respect to the classification model $f(\boldsymbol{x}; \theta)$ is

$$R(f) = \mathbb{E}_{(\boldsymbol{x}, y)}[\mathcal{L}(f(\boldsymbol{x}), y)].$$

Let $\widehat{R}(f) = \frac{1}{n}\sum_{i=1}^{n}\mathcal{L}(f(\boldsymbol{x_i}), y_i)$ denote the corresponding empirical risk. However, in NPLL, we cannot minimize the empirical risk directly as the ground-truth label $y$ is inaccessible. Therefore, we need to train the model with $\widehat{R}'(f) = \frac{1}{n}\sum_{i=1}^{n}\mathcal{L}_{PLL}(f(\boldsymbol{x_i}), Y_i)$, where $Y_i$ denotes the CLS of the instance $\boldsymbol{x_i}$. Let $\hat{f} = \arg\min_{f\in\mathcal{F}}\widehat{R}'(f)$ be the empirical risk minimizer, and $f^* = \arg\min_{f\in\mathcal{F}}R(f)$ be the true risk minimizer. Let $\mathcal{L}_{PLL}(f(\boldsymbol{x_i}), Y_i) = \frac{1}{|Y_i|}\sum_{c\in Y_i}\mathcal{L}(f(\boldsymbol{x_i}), c)$ be the loss for PLL. Let $n' = \sum_{i=1}^{n}I(y_i \in Y_i)$ be the number of normal samples. Let $\epsilon$ represent the noise rate in the NPLL dataset which is defined as the proportion of instances whose ground-truth label does not belong to CLS, i.e., $\epsilon = \frac{n-n'}{n}$ and let $\alpha = \frac{1}{n'}\sum_{i=1,y_i\in Y_i}^{n}|Y_i|$ be the mean size of CLS for normal samples. Besides, we define the function space $\mathcal{H}_y$ for the label $y \in \mathcal{Y}$ as $\{h : \boldsymbol{x} \mapsto f_y(\boldsymbol{x})|f \in \mathcal{F}\}$ where $f_y(\boldsymbol{x})$ represents the predicted probability of the $y$-th class for $\boldsymbol{x}$. Let $\mathfrak{R}_n(\mathcal{H}_y)$ be the expected Rademacher complexity (Bartlett & Mendelson, 2002) of $\mathcal{H}_y$ with sample size $n$. Then we have the following theorem.

**Theorem 1.** *Assume the loss function $\mathcal{L}(f(\boldsymbol{x}), y)$ is $\rho$-Lipschitz with respect to $f(\boldsymbol{x})$ for all $y \in \mathcal{Y}$ and uppper-bounded by $M$. For noise rate $0 < \epsilon < 1$ and mean CLS size for normal samples $1 < \alpha < C$, for any $\delta > 0$, with probability at least $1 - \delta$, we have*

$$R(\hat{f}) - R(f^*) \leq 2(1 - \frac{1-\epsilon}{\alpha})M + 4\sqrt{2}\rho\sum_{y=1}^{C}\mathfrak{R}_n(\mathcal{H}_y) + 2M\sqrt{\frac{log\frac{2}{\delta}}{2n}}.$$

The proof of Theorem 1 is provided in Appendix A.1. It can be observed that the generalization performance of $\hat{f}$ is primarily influenced by three factors: the noise rate $\epsilon$, the mean CLS size $\alpha$ of normal samples, and the sample size $n$. As $n \to \infty$, $\epsilon \to 0$ and $\alpha \to 1$, Theorem 1 shows that the generalized error bound will be reduced, and the empirical risk minimizer $\hat{f}$ will get closer to the true risk minimizer $f^*$. **Obviously, a smaller noise rate $\epsilon$ and a smaller CLS size $\alpha$ will bring better generalization performance.**

## 4 PROPOSED METHOD

Based on the theoretical analyses above, we can find that the key to solving the NPLL problem lies in reducing the noise rate in the dataset and minimizing the size of the CLS. To achieve these goals, our method comprises two components. First, we try to separate the samples into highly reliable normal samples, highly reliable noisy samples, and uncertain samples. Second, based on the above sample separation, we separately perform CLS reconstruction with an instance-adaptive parameter for these samples. As the model is trained on reconstructed CLS, its performance improves progressively, which in turn promotes the aforementioned processes, ultimately leading to a reduction in noise rate while simultaneously decreasing the size of the CLS. The pseudo-code is summarized in Algorithm 1. The framework of our method is showed in Fig. 4.

### 4.1 PROGRESSIVE SAMPLE SEPARATION

We first propose a metric, the consistency **e**rror between **C**LS-based and **K**NN-based pseudo-label (ECK), to distinguish normal samples and noisy samples. Considering that the model output is not reliable in the early stages of training for some hard samples, **we divide the samples into three parts based on the ECK metric**: highly reliable normal samples, highly reliable noisy samples, and uncertain samples. As the performance of model improves, we gradually increase the proportion of highly reliable samples to achieve an accurate separation between normal samples and noisy samples.

According to the smooth assumption (Wang et al., 2022a), similar samples tend to share the same label. For each sample $\boldsymbol{x_i}$, we select its $K$-nearest neighbor samples $N_i$ with cosine distance to construct a **KNN-based pseudo-label**:

$$\boldsymbol{q_i} = Normalize\left(\sum_{j\in N_i}s_{ij}\cdot\boldsymbol{p_j}\right), \tag{1}$$

where $s_{ij}$ is cosine similarity of features between sample $\boldsymbol{x_i}$ and $\boldsymbol{x_j}$, $\boldsymbol{p_j} = f(\boldsymbol{x_j}; \theta)$ is the probability prediction vector of $\boldsymbol{x_j}$, and $Normalize(\cdot)$ is a normalization function that ensures the sum of vector

---

**Algorithm 1** Training Process of the Proposed Method

---

**Input**: NPLL training dataset $\mathcal{D} = \{(\boldsymbol{x_i}, Y_i)\}_{i=1}^{n}$, classifier $f$, PLL method $\mathcal{L}_{PLL}$, warm-up epoch $e_w$, uncertainty-end epoch $e_{end}$, number of epoch $e_{max}$ hyper-parameters $\lambda$, $\beta$.
**Output**: The optimized muti-class classifier $f$.

1: **for** $epoch = 1, 2, \cdots, e_w$ **do**
2:     Warm-up training $f$ by $\mathcal{L}_{PLL}$ with Dataset $\mathcal{D} = \{(\boldsymbol{x_i}, Y_i)\}_{i=1}^{n}$
3: **end for**
4: **for** $epoch = 1, 2, \cdots, e_{max}$ **do**
5:     Calculate percentage $r_l$ and $r_u$ by Eq. (5)
6:     Calculate ECK $E_i$ for each instance $\boldsymbol{x_i}$ by Eq. (3)
7:     $u \leftarrow E^{r_u}$ // $E^{r_u}$ is $r_u$-percentile of $E_i$
8:     $l \leftarrow E^{r_l}$ // $E^{r_l}$ is $r_l$-percentile of $E_i$
9:     Calculate $v_i$ for each instance $\boldsymbol{x_i}$ by Eq. (4) // Sample Separation
10:     Calculate $\widehat{Y}$ by Eqs. (8) (9) (10) // CLS Reconstruction
11:     Train $f$ by $\mathcal{L}_{PLL}$ with Dataset $\mathcal{D} = \{(\boldsymbol{x_i}, \widehat{Y}_i)\}_{i=1}^{n}$
12: **end for**

---

equal to 1. For each sample $\boldsymbol{x_i}$, we further construct a **CLS-based pseudo label** by normalizing the model's prediction to the candidate labels:

$$\widetilde{\boldsymbol{q_i}} = Normalize\left(\boldsymbol{p_i} \odot \boldsymbol{S}(Y_i)\right), \tag{2}$$

where $\boldsymbol{S}(\cdot)$ transfers CLS to a vector where the $i$-th element equal to 1 if the $i$-th label is in the CLS, and 0 otherwise. Operation $\odot$ is Hadamard product, i.e., element-wise multiplication. Then, we calculate the ECK of each sample $\boldsymbol{x_i}$ using the cross entropy between $\widetilde{\boldsymbol{q_i}}$ and $\boldsymbol{q_i}$, i.e.,

$$E_i = -\sum_{j=1}^{C} \widetilde{q}_{ij} \log q_{ij}, \tag{3}$$

where $\widetilde{q}_{ij}$ and $q_{ij}$ are the $j$-th elements of the vectors $\widetilde{\boldsymbol{q_i}}$ and $\boldsymbol{q_i}$, respectively. When the ground-truth label of a sample is included in the candidate labels, the CLS-based pseudo-labels and KNN-based pseudo-labels tend to be similar, resulting in a relatively small value of ECK. However, when the the ground-truth label of a sample locates at the non-candidate labels, there will be a significant discrepancy between the CLS-based pseudo-label and KNN-based pseudo-label, leading to a larger value of ECK.

We employ a dual-threshold approach to separate the samples into three distinct groups: highly reliable normal samples, highly reliable noisy samples, and uncertain samples. Then, we use a ternary variable $v_i$ to denote the group to which the $i$-th sample $\boldsymbol{x_i}$ belongs. Specifically, when $v_i$ is 1, 0, and $-1$, it indicates that the $i$-th sample belongs to the normal samples, uncertain samples, and noisy samples, respectively. Consequently, we have

$$v_i = \begin{cases} 1, & E_i \leq l, \\ 0, & l < E_i \leq u, \\ -1, & E_i > u, \end{cases} \tag{4}$$

where the thresholds $l$ and $u$ are used to balance the quality and quantity of sample separation. As shown in Fig. 4, with model training, the value of $l$ increases while the value of $u$ decreases, until $l$ equals $u$ all samples will be separated into noisy samples and normal samples. In practice, as the ECK changes across different epochs, $l$ and $u$ are assigned the $r_l$ and $r_u$ percentiles of ECK, respectively, to ensure stability during training. Due to the typically lower proportion of noisy samples compared to normal ones, we employ the hyper-parameter $\lambda = \frac{|\Delta r_u|}{|\Delta r_l|}, (0 < \lambda \leq 1)$ to regulate the rate of percentile change between $r_l$ and $r_u$. We linearly ramp up $r_l$ and taper off $r_u$, i.e.,

$$r_l = \frac{1}{1+\lambda} * \max(\frac{epoch - e_w}{e_{end} - e_w}, 1), \quad r_u = 1 - \frac{\lambda}{1+\lambda} * \max(\frac{epoch - e_w}{e_{end} - e_w}, 1) \tag{5}$$

where $epoch$ is the current training epoch, $e_w$ is the warm up epoch and the $e_{end}$ denotes the epoch where all samples are separated, i.e., $r_l = r_u$. So far, we have partitioned the samples into three

groups: normal samples $\{\boldsymbol{x_i}|v_i = 1, \forall i\}$, noisy samples $\{\boldsymbol{x_i}|v_i = -1, \forall i\}$, and uncertain samples $\{\boldsymbol{x_i}|v_i = 0, \forall i\}$. As the model iterates, the model is better trained and accordingly the number of uncertain samples will gradually decrease.

**Difference with the previous noisy sample separation methods.** The previous methods, PiCO+ (Wang et al., 2024) and UPLLRS (Shi et al., 2023), also employ a partitioning strategy for noisy samples. However, different with our method, they overlook the existence of the uncertain samples during the training process. Directly partitioning uncertain samples as noisy samples or normal ones will inevitably lead to more errors, especially in the early stages of model training when the model's capability is weak. Our proposed dual-threshold strategy gradually increases the partitioning degree as the model's capability improves, and the experimental results in Table 5 and Table 7 demonstrate its superiority over the previous methods. Additionally, these methods just treat the partitioned noisy samples as unlabeled samples, ignoring the prior that ground-truth label of noisy samples locates in the non-CLS. In contrast, the CLS reconstruction method we propose in the next section will utilize this important prior.

## 4.2 RECONSTRUCTION OF CANDIDATE LABEL SET

In the previous section, we have identified a subset of normal and noisy samples. A straightforward approach to handle noisy samples is treating their non-CLS as their new CLS (ground-truth must locate at non-CLS for noisy samples). Although this approach can reduce the noise rate, it will increase the size of CLS for samples (the size of CLS is usually smaller than that of the non-CLS), thereby violating the conclusion in Theorem 1. Therefore, we propose to reconstruct the CLS for all instances, which aims to reduce the size of the CLS while ensuring that the ground-truth label is included in the CLS. To achieve those two goals simultaneously, we formulate the following optimization problem :

$$\min_{\widehat{Y}} \quad \sum_{i=1}^{n} I(v_i \neq 0) \left( |\widehat{Y}_i| - \beta \langle \boldsymbol{q_i}, \boldsymbol{S}(\widehat{Y}_i) \rangle \right), \tag{6}$$
$$\text{s.t.} \quad \forall i, \text{if } v_i = 1, \widehat{Y}_i \neq \varnothing, \widehat{Y}_i \subseteq Y_i,$$
$$\forall i, \text{if } v_i = -1, \widehat{Y}_i \neq \varnothing, \widehat{Y}_i \subseteq \mathcal{Y} - Y_i,$$

where $\widehat{Y}_i$ is reconstructed CLS of $\boldsymbol{x_i}$ and $\boldsymbol{S}(\widehat{Y}_i)$ transfers the reconstructed CLS $\widehat{Y}_i$ to a vector of that the $j$-th element equal to 1 if the label $j$ is in $\widehat{Y}_i$, and 0 otherwise. $|\cdot|$ is the cardinality of a set and $\langle \cdot, \cdot \rangle$ denote the inner product of two vectors. $I(\cdot)$ is an indicator function which outputs 1 if the condition holds, 0 otherwise. $\boldsymbol{q_i}$ is the KNN-based pseudo label of $\boldsymbol{x_i}$ obtained by Eq. (1).

Minimizing $|\widehat{Y}_i|$ in the optimization objective aims to reduce the CLS size, while minimizing $-\langle \boldsymbol{q_i}, \boldsymbol{S}(\widehat{Y}_i) \rangle$ is designed to select more reliable labels to reconstruct the CLS. The hyper-parameter $\beta$ balances these two objectives. The solution is showed on Appendix A.2, i.e., $\widehat{Y}_i = \{j|j \in Y_i, q_{ij} > \frac{1}{\beta}\}$ if $\boldsymbol{x_i}$ is a normal sample ($v_i = 1$) while $\widehat{Y}_i = \{j|j \in \mathcal{Y} - Y_i, q_{ij} > \frac{1}{\beta}\}$ if $\boldsymbol{x_i}$ is a noisy sample ($v_i = -1$). Considering the various learning states of samples, for samples with high confidence, we can intensify the reduction of CLS with a lower value of $\beta$. Conversely, for samples with low confidence, it is appropriate to set higher $\beta$ to ensure the ground-truth label within the CLS. Therefore, we transform the hyper-parameter $\beta$ into an instance-adaptive parameter. Specifically, for the sample $\boldsymbol{x_i}$, the instance-adaptive parameter is

$$\beta_i = \tau(\boldsymbol{q_i}), \tag{7}$$

where $\tau(\cdot)$ is a function, and the output of $\tau(\cdot)$ is inversely proportional to the pseudo-label confidence. In practice, we use the maximum value in the pseudo-label to represent the confidence of the sample, i.e. $\tau(\boldsymbol{q_i}) = \frac{\beta}{max(\boldsymbol{q_i})}$, where $\beta$ is the hyper-parameter in Eq. (6). Finally, the reconstructed CLS is as follows:

$$\widehat{Y}_i = \{j|j \in Y_i, \ q_{ij} > \frac{1}{\beta_i}\}, \quad \text{if } v_i = 1, \tag{8}$$

$$\widehat{Y}_i = \{j|j \in \mathcal{Y} - Y_i, \ q_{ij} > \frac{1}{\beta_i}\}, \quad \text{if } v_i = -1. \tag{9}$$

In the early stages of training, there are fewer selected reliable samples. Directly excluding uncertain samples $\{\boldsymbol{x_i}|v_i = 0, \forall i\}$ from training will slow down the convergence to some extent. Therefore, for these uncertain samples, we choose to add the label with the highest pseudo-label probability from the non-CLS to the CLS, and remove the label with the lowest pseudo-label probability from the CLS, i.e.,

$$\widehat{Y}_i = Y_i \cup \{c\} \setminus \{c'\}, \quad \text{if } v_i = 0, \tag{10}$$

where $c = \arg\max_{j \in \mathcal{Y} - Y_i} q_{ij}$ and $c' = \arg\min_{j \in Y_i} q_{ij}$.

**Remark.** Driven by theoretical analyses in Section 3, we propose a CLS reconstruction method that corrects noisy labels while reducing the size of the CLS. Note that all the previous methods do not consider the impact of CLS size on generalization performance. The experimental results in Table 5 validate the effectiveness of the proposed reconstruction method.

### 4.3 ITERATIVE IMPROVEMENT

Utilizing the CLS-reconstructed dataset obtained from the previous section, we optimize the model by

$$\min_{\theta} \quad \frac{1}{n} \sum_{i=1}^{n} \mathcal{L}_{PLL}(f(\boldsymbol{x_i}; \theta), \widehat{Y}_i), \tag{11}$$

where $\mathcal{L}_{PLL}$ can be any PLL method. Eq. (11) can be solved by applying the stochastic gradient decent (SGD) method. As the performance of the model improves, it will in turn enhance the effect of sample separation and CLS reconstruction. By alternating optimization, the model thus achieves the excellent performance on NPLL dataset. As $\mathcal{L}_{PLL}$ can be any PLL method, our approach serves as a plug-in to enhance the performance of traditional PLL methods on NPLL datasets.

## 5 EXPERIMENTS

### 5.1 EXPERIMENTAL SETUP

**Datasets.** Following the previous works (Wang et al., 2024; Xu et al., 2023; Qiao et al., 2023), we first evaluated our method on two benchmark datasets, CIFAR10 and CIFAR100 (Krizhevsky et al., 2009). Following the same dataset generation process of the previous works (Wang et al., 2024), for each benchmark dataset, we manually corrupted its training dataset into NPLL dataset. Specifically, we first constructed the candidate label set for each sample by manually flipping incorrect labels $\overline{y} \neq y$ to candidate labels with probability $\eta = P(\overline{y} \in Y_i | \overline{y} \neq y)$. Then, each sample has a probability $\gamma$ of being the noisy sample, i.e., the probability of that CLS do not contain ground-truth label $y$ is $\gamma$. We denote the probability $\eta$ as the ambiguity level and the probability $\gamma$ as the noise level. We consider $\eta \in \{0.3, 0.4, 0.5\}$ for CIFAR10 and $\eta \in \{0.03, 0.05, 0.1\}$ for CIFAR100. We choose noise level $\gamma \in \{0.2, 0.3, 0.4\}$. Considering that similar categories are more likely to be added to the candidate set, partial label annotations are prone to arise from fine-grained image scenarios. We conducted experiments on three fine-grained datasets CIFAR100H, CUB200 (Welinder et al., 2010) and Flower (Nilsback & Zisserman, 2008). We further evaluated our method on two real-world crowdsourced datasets Treeversity and Benthic (Schmarje et al., 2022) to confirm the effectiveness of the our method in real-world applications. The detailed information of these datasets is presented in Appendix A.6.

**Baselines.** We compared our method with four recent PLL methods including PRODEN (Lv et al., 2020), CC (Feng et al., 2020), CRDPLL (Wu et al., 2022), and PaPi (Xia et al., 2023) and four SOTA NPLL methods including FREDIS (Qiao et al., 2023), PiCO+ (Wang et al., 2024) and ALIM (Xu et al., 2023) (two variants: ALIM-Onehot and ALIM-Scale). It is noteworthy that ALIM can also serve as a plug-in applied to other PLL methods.

**Implementation Details.** Following the experimental setup (Xu et al., 2023), we splited a clean validation set from the training set to determine hyper-parameters. Then, we transformed the validation set back to its NPLL form and incorporated them into the training set to retrain the model. Due to our approach is a plug-in method, we default to combining our method with PaPi (Xia et al., 2023) for NPLL unless specifically stated. More experimental details are showed in Appendix A.3.

Table 1: Accuracy comparisons on CIFAR10 and CIFAR100 under various ambiguity levels $\eta$ and noise levels $\gamma$. Bold indicates the best result. Accuracies are presented in percentage (%) form. All experiments were conducted three times under the same three distinct random seeds.

| Method | CIFAR10 | | | | | | | | |
|---|---|---|---|---|---|---|---|---|---|
| | $\eta = 0.3$ | | | $\eta = 0.4$ | | | $\eta = 0.5$ | | |
| | $\gamma = 0.2$ | $\gamma = 0.3$ | $\gamma = 0.4$ | $\gamma = 0.2$ | $\gamma = 0.3$ | $\gamma = 0.4$ | $\gamma = 0.2$ | $\gamma = 0.3$ | $\gamma = 0.4$ |
| PRODEN [ICML'20] | 77.74 ± 0.53 | 67.20 ± 0.99 | 57.74 ± 0.56 | 71.43 ± 0.54 | 59.28 ± 0.82 | 46.87 ± 1.40 | 63.94 ± 0.75 | 49.38 ± 1.13 | 32.03 ± 1.33 |
| CC [NeurIPS'20] | 75.09 ± 0.37 | 63.48 ± 1.72 | 54.42 ± 0.34 | 68.08 ± 0.94 | 54.46 ± 0.36 | 42.24 ± 1.31 | 58.22 ± 0.24 | 44.38 ± 1.60 | 28.57 ± 1.67 |
| CRDPLL [IMCL'23] | 84.61 ± 0.19 | 80.12 ± 0.46 | 71.43 ± 0.93 | 81.60 ± 0.46 | 72.79 ± 0.39 | 53.24 ± 2.30 | 76.92 ± 1.04 | 56.78 ± 0.76 | 32.60 ± 1.04 |
| PaPi [CVPR'23] | 89.80 ± 0.36 | 86.36 ± 1.06 | 78.45 ± 0.61 | 86.71 ± 0.65 | 81.78 ± 0.52 | 59.02 ± 1.67 | 86.34 ± 0.67 | 73.06 ± 1.16 | 47.16 ± 1.35 |
| FREDIS [ICML'23] | 92.09 ± 0.29 | 87.91 ± 1.74 | 84.15 ± 0.19 | 89.25 ± 2.18 | 84.78 ± 2.50 | 77.74 ± 0.70 | 88.10 ± 0.59 | 79.73 ± 2.70 | 52.68 ± 1.22 |
| PiCO+ [TPAMI'24] | 94.12 ± 0.35 | 94.22 ± 1.19 | 89.56 ± 0.52 | 93.84 ± 0.96 | 92.96 ± 0.92 | 85.94 ± 1.48 | 92.21 ± 0.66 | 89.63 ± 1.47 | 75.59 ± 1.32 |
| ALIM-Scale [NeurIPS'23] | 94.97 ± 0.27 | 94.10 ± 0.16 | 93.31 ± 0.27 | 94.38 ± 0.24 | 93.41 ± 0.25 | 89.62 ± 0.66 | 93.92 ± 0.09 | 90.10 ± 0.56 | 69.78 ± 1.07 |
| ALIM-Onehot [NeurIPS'23] | 95.13 ± 0.10 | 94.39 ± 0.23 | 93.68 ± 0.14 | 94.64 ± 0.08 | 94.17 ± 0.04 | 88.88 ± 0.30 | 94.07 ± 0.15 | 90.71 ± 0.73 | 65.57 ± 2.04 |
| Ours | **96.91 ± 0.17** | **96.80 ± 0.14** | **96.47 ± 0.19** | **96.78 ± 0.10** | **96.23 ± 0.66** | **96.03 ± 0.56** | **96.55 ± 0.02** | **94.54 ± 1.84** | **82.63 ± 1.70** |

| Method | CIFAR100 | | | | | | | | |
|---|---|---|---|---|---|---|---|---|---|
| | $\eta = 0.03$ | | | $\eta = 0.05$ | | | $\eta = 0.1$ | | |
| | $\gamma = 0.2$ | $\gamma = 0.3$ | $\gamma = 0.4$ | $\gamma = 0.2$ | $\gamma = 0.3$ | $\gamma = 0.4$ | $\gamma = 0.2$ | $\gamma = 0.3$ | $\gamma = 0.4$ |
| PRODEN [ICML'20] | 57.83 ± 0.49 | 48.66 ± 0.31 | 40.10 ± 0.37 | 55.39 ± 0.61 | 45.36 ± 1.16 | 36.11 ± 0.40 | 50.88 ± 1.12 | 40.02 ± 1.40 | 28.81 ± 0.89 |
| CC [NeurIPS'20] | 57.73 ± 0.70 | 48.66 ± 0.28 | 38.26 ± 1.31 | 55.93 ± 0.70 | 45.41 ± 1.23 | 35.31 ± 0.07 | 51.81 ± 0.36 | 40.69 ± 0.65 | 28.56 ± 0.29 |
| CRDPLL [IMCL'23] | 63.91 ± 0.53 | 59.16 ± 0.14 | 55.16 ± 0.36 | 63.02 ± 0.52 | 57.77 ± 0.48 | 53.64 ± 0.29 | 61.43 ± 0.21 | 54.77 ± 0.05 | 48.50 ± 0.36 |
| PaPi [CVPR'23] | 69.83 ± 0.57 | 61.99 ± 0.24 | 59.71 ± 0.68 | 68.64 ± 0.61 | 62.72 ± 0.95 | 58.63 ± 0.25 | 67.64 ± 0.56 | 61.98 ± 0.70 | 55.60 ± 0.51 |
| FREDIS [ICML'23] | 66.94 ± 0.10 | 61.85 ± 0.41 | 57.99 ± 0.35 | 67.48 ± 0.57 | 62.72 ± 0.77 | 57.19 ± 0.68 | 66.09 ± 0.42 | 57.60 ± 0.64 | 45.09 ± 0.72 |
| PiCO+ [TPAMI'24] | 74.32 ± 0.43 | 72.68 ± 0.28 | 67.31 ± 0.58 | 73.33 ± 0.48 | 70.17 ± 0.62 | 65.01 ± 0.48 | 62.67 ± 0.46 | 56.25 ± 0.84 | 47.75 ± 1.08 |
| ALIM-Scale [NeurIPS'23] | 76.39 ± 0.71 | 75.40 ± 0.60 | 74.58 ± 0.25 | 76.02 ± 0.31 | 75.33 ± 0.14 | 74.49 ± 0.69 | 75.27 ± 0.22 | 71.06 ± 1.41 | 64.61 ± 2.37 |
| ALIM-Onehot [NeurIPS'23] | 76.29 ± 0.19 | 74.83 ± 0.12 | 73.39 ± 1.14 | 74.92 ± 0.48 | 74.40 ± 0.06 | 71.49 ± 1.02 | 61.24 ± 0.57 | 58.01 ± 1.03 | 47.27 ± 1.82 |
| Ours | **81.74 ± 0.16** | **80.73 ± 0.16** | **79.95 ± 0.20** | **80.76 ± 0.08** | **80.17 ± 0.20** | **78.89 ± 0.41** | **79.98 ± 0.23** | **79.21 ± 0.88** | **76.18 ± 1.67** |

Table 2: Accuracy comparisons when the methods are used as a plug-in on CIFAR10 and CIFAR100 under various ambiguity levels $\eta$ and noise levels $\gamma$. Bold indicates the best result. Accuracies are presented in percentage (%) form.

| Method | CIFAR10 | | | | | | | | |
|---|---|---|---|---|---|---|---|---|---|
| | $\eta = 0.3$ | | | $\eta = 0.4$ | | | $\eta = 0.5$ | | |
| | $\gamma = 0.2$ | $\gamma = 0.3$ | $\gamma = 0.4$ | $\gamma = 0.2$ | $\gamma = 0.3$ | $\gamma = 0.4$ | $\gamma = 0.2$ | $\gamma = 0.3$ | $\gamma = 0.4$ |
| PRODEN | 78.00 | 67.57 | 57.75 | 71.31 | 60.22 | 48.38 | 64.62 | 49.95 | 31.93 |
| PRODEN + ALIM-Onehot | 90.83 | 88.64 | 84.87 | 89.15 | 84.95 | 77.71 | 86.63 | 79.89 | 42.83 |
| PRODEN + ALIM-Scale | 92.05 | 89.83 | 83.22 | 90.58 | 85.78 | 71.27 | 87.10 | 66.34 | 38.14 |
| PRODEN + Ours | **94.35** | **94.10** | **93.30** | **94.21** | **93.80** | **90.48** | **94.00** | **93.27** | **62.29** |
| CRDPLL | 84.40 | 79.61 | 71.46 | 81.97 | 72.43 | 55.06 | 76.93 | 56.40 | 31.96 |
| CRDPLL + ALIM-Onehot | 88.30 | 83.64 | 74.21 | 86.12 | 77.04 | 56.58 | 80.75 | 60.72 | 32.95 |
| CRDPLL + ALIM-Scale | 92.06 | 90.42 | 85.86 | 90.81 | 85.36 | 73.85 | 87.06 | 68.57 | 40.50 |
| CRDPLL + Ours | **95.29** | **95.25** | **94.86** | **95.17** | **94.44** | **90.43** | **94.28** | **83.40** | **54.66** |
| PaPi | 69.83 | 61.99 | 59.71 | 68.64 | 62.72 | 58.63 | 67.64 | 61.98 | 55.60 |
| PaPi + ALIM-Onehot | 95.64 | 94.88 | 92.57 | 95.67 | 94.26 | 91.01 | 94.45 | 90.50 | 66.50 |
| PaPi + ALIM-Scale | 95.11 | 94.40 | 92.32 | 95.94 | 94.15 | 90.25 | 93.24 | 88.41 | 58.92 |
| PaPi + Ours | **96.74** | **96.67** | **96.41** | **96.84** | **95.47** | **95.39** | **96.53** | **95.41** | **81.85** |

| Method | CIFAR100 | | | | | | | | |
|---|---|---|---|---|---|---|---|---|---|
| | $\eta = 0.03$ | | | $\eta = 0.05$ | | | $\eta = 0.1$ | | |
| | $\gamma = 0.2$ | $\gamma = 0.3$ | $\gamma = 0.4$ | $\gamma = 0.2$ | $\gamma = 0.3$ | $\gamma = 0.4$ | $\gamma = 0.2$ | $\gamma = 0.3$ | $\gamma = 0.4$ |
| PRODEN | 58.10 | 48.98 | 40.30 | 54.89 | 46.60 | 35.85 | 51.87 | 41.63 | 29.84 |
| PRODEN + ALIM-Onehot | 74.07 | 71.18 | 68.26 | 72.47 | 69.76 | 66.36 | 68.17 | 62.33 | 53.44 |
| PRODEN + ALIM-Scale | 74.98 | 73.50 | 69.05 | 74.49 | 72.14 | 66.88 | 70.64 | 64.17 | 55.10 |
| PRODEN + Ours | **75.66** | **74.59** | **72.29** | **75.08** | **73.68** | **70.19** | **74.74** | **69.93** | **61.47** |
| CRDPLL | 64.36 | 59.01 | 55.20 | 62.45 | 58.26 | 53.37 | 61.26 | 54.72 | 48.69 |
| CRDPLL + ALIM-Onehot | 68.31 | 63.70 | 57.20 | 67.21 | 64.24 | 55.37 | 67.09 | 62.28 | 50.19 |
| CRDPLL + ALIM-Scale | 70.93 | 67.99 | 58.60 | 70.09 | 67.30 | 57.43 | 68.24 | 63.03 | 53.74 |
| CRDPLL + Ours | **74.46** | **72.93** | **69.96** | **73.58** | **72.39** | **68.19** | **72.27** | **69.55** | **66.16** |
| PaPi | 69.83 | 61.99 | 59.71 | 68.64 | 62.72 | 58.63 | 67.64 | 61.98 | 55.60 |
| PaPi + ALIM-Onehot | 80.37 | 79.20 | 77.58 | 79.64 | 78.14 | 76.08 | 77.77 | 74.04 | 58.63 |
| PaPi + ALIM-Scale | 81.50 | 80.23 | 78.77 | 80.51 | 78.79 | 77.11 | 79.16 | 75.86 | 63.02 |
| PaPi + Ours | **81.70** | **80.69** | **80.08** | **80.85** | **79.88** | **79.29** | **79.78** | **78.23** | **78.10** |

## 5.2 MAIN RESULTS

**Our method achieves SOTA classification accuracy.** As shown in Table 1, on the CIFAR10 and CIFAR100 datasets with different ambiguity levels $\eta$ and different noise levels $\gamma$, our method outperforms all the compared methods by a large margin. For example, on CIFAR100 with $\eta = 0.1$, our method improves the best compared method by **4.71%**, **8.15%** and **11.57%** on three different noise levels $\gamma = 0.2, 0.3$ and $0.4$ respectively. On CIFAR10 with $\eta = 0.5$, our method performs **2.48%**, **3.83%** and **7.04%** better than the best compared method.

**Our method can promote the performance of different PLL methods on NPLL dataset.** Since our method is a plug-in approach, we combine it with various existing PLL methods and compare the

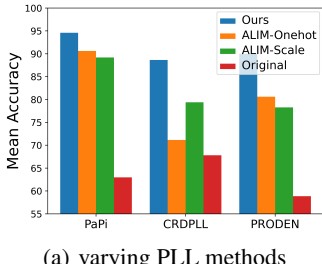 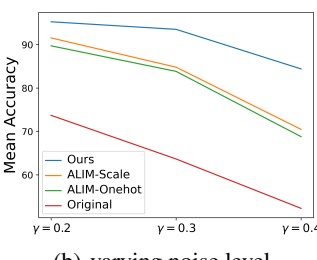 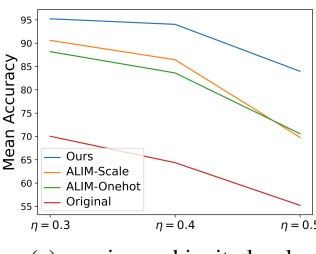

(a) varying PLL methods     (b) varying noise level     (c) varying ambiguity level

Figure 1: The mean accuracy on CIFAR10 of our method and the ALIM across different PLL methods, ambiguity levels, and noise levels. The results are derived from the statistics presented in Table 2. The results on CIFAR100 is showed in Appendix A.5.

results with that of ALIM (Xu et al., 2023) in Table 2. It can be observed that our method significantly enhances the performance of different PLL methods across various ambiguity levels and noise levels and outperforms ALIM in all cases. For example, on CIFAR100 ($\eta = 0.1$, $\gamma = 0.4$), integrated with PRODEN, CRDPLL and PaPi, our method performs **6.37%**, **12.42%** and **15.08%** better than the ALIM. Fig. 1 shows the mean accuracy of our method and that of ALIM across different PLL methods, ambiguity levels, and noise levels. As the difficulty of the problem increases (with higher levels of noise and ambiguity), our method demonstrates a more pronounced enhancement in the performance of the PLL model compared to the current SOTA method.

**Our method remains effectiveness on fine-grained datasets for NPLL task.** Partial label annotations are easily arisen from fine-grained image scenarios as annotators are more likely to confuse similar categories. Therefore we further conduct experiments on fine-grained datasets, CIFAR100H (Krizhevsky et al., 2009), CUB200 (Welinder et al., 2010) and Flower (Nilsback & Zisserman, 2008). The results in Table 3 demonstrate that our method significantly improves

Table 3: Accuracies (%) on fine-grained datasets.

| Method | CIFAR100H $\eta = 0.5$ $\gamma = 0.2$ | CUB-200 $\eta = 0.03$ $\gamma = 0.3$ | Flower $\eta = 0.05$ $\gamma = 0.2$ |
|---|---|---|---|
| PaPi | 63.94 | 43.56 | 74.95 |
| PaPi + ALIM-Onehot | 69.29 | 48.58 | 76.07 |
| PaPi + ALIM-Scale | 74.34 | 51.44 | 78.47 |
| PaPi + Ours | **76.93** | **52.78** | **81.72** |

PLL method PaPi on fine-grained NPLL task by **6.77%**, **9.22%** and **12.99%** on Flower ($\eta = 0.05$, $\gamma = 0.2$), CUB200 ($\eta = 0.03$, $\gamma = 0.3$) and CIFAR100H ($\eta = 0.5$, $\gamma = 0.2$). And our method, as expected, also outperforms the current SOTA NPLL method ALIM. This results further substantiate the efficacy of our method.

**Our method is skilled in real-world datasets for NPLL task.** To further verify the superiority of our method, we evaluated different methods on real-world crowdsourced datasets: Treeversity and Benthic. The details of those datasets and the experimental settings are showed in Appendix A.6. As indicated in Table 4, our method still significantly outperforms the current SOTA methods in realistic experiment.

Table 4: Accuracies (%) on real-world datasets.

| Method\Dataset | Treeversity2 | Treeversity3 | Benthic2# |
|---|---|---|---|
| Papi | 81.07 | 82.55 | 80.90 |
| PaPi + ALIM-Scale | 82.72 | 83.47 | 81.46 |
| PaPi + ALIM-Onehot | 84.54 | 86.01 | 82.24 |
| PaPi + Ours | **86.41** | **86.67** | **83.47** |

Especially, in more challenging scenarios like Treeversity2, the performance gap is even more significant.

**Our method is capable of effectively segregating the reliable samples.** We validated the rationality of the progressive sample separation in Fig. 2 which visualizes the distribution of ECK and the boundary values $l$ and $u$. It can be observed that as the training epoch increases, ECK increasingly distinguishes between normal samples and noisy samples. Meanwhile, the boundary values $l$ and $u$ can effectively separate highly reliable normal samples from highly reliable noisy samples. Moreover, the number of the highly reliable samples gradually increases with the training of the model until all samples are selected. Fig. 3(a) illustrates the trends in separation accuracy and the number of selected highly reliable samples as the model training, where we find that with the model training, the highly reliable selection quantities of both normal and noisy samples increase, while the quality maintains at a high level. We further evaluated separation accuracy of normal and noisy samples for different NPLL method in Appendix A.7. We can observe that our method outperforms other NPLL

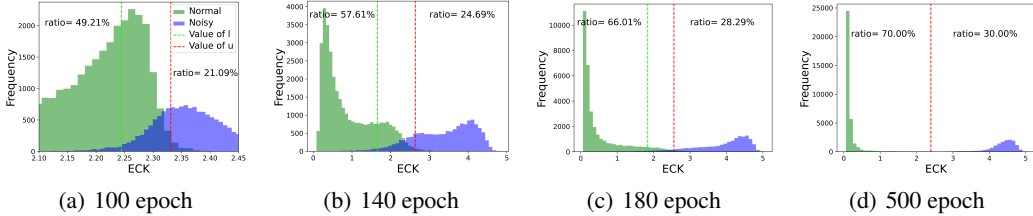

| (a) 100 epoch | (b) 140 epoch | (c) 180 epoch | (d) 500 epoch |

Figure 2: Distribution of the ECK in Eq. (3) for real normal and noisy samples with increasing training iterations. The experiment is conducted on CIFAR10 ( $\eta = 0.5$, $\gamma = 0.3$ ). The term "ratio" on the graph represents the proportion of each selected subset of samples relative to the total number of samples. The results on CIFAR100 are showed are Appendix A.5.

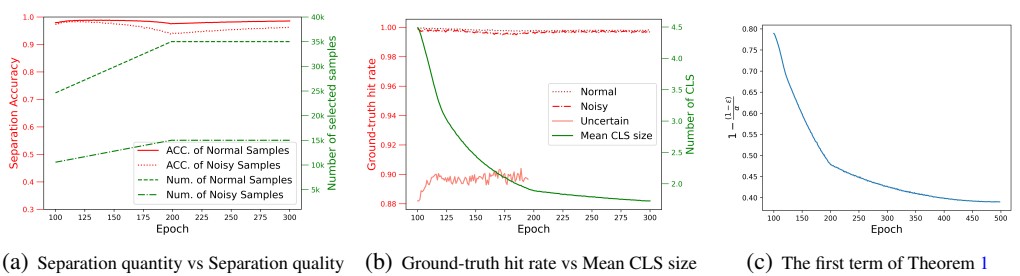

(a) Separation quantity vs Separation quality  (b) Ground-truth hit rate vs Mean CLS size  (c) The first term of Theorem 1

Figure 3: The experiment is conducted on CIFAR10 ($\eta = 0.5$, $\gamma = 0.3$). (a) demonstrates that with the progression of training iterations, the selection quantities of both normal and noisy samples increase, while the quality (accuracy) is able to be sustained at a high level. (b) reflects that, as training iterations continue, the reconstructed CLS size consistently decreases (green curve) while ensuring a high ground-truth label hit rate (the rate of samples that ground-truth label locates in the reconstructed CLS). There are no uncertain samples after 200 epoch, because we set $e_{end} = 200$. (c) indicates that, as the model training progresses, the value of $1 - \frac{1-\epsilon}{\alpha}$ is continuously decreasing. Note this value is positively related to generalization error bound in Theorem 1. The results on CIFAR100 are showed in Appendix A.5.

methods with a substantial gap especially when the noise level is high, which validates the superiority of ECK and dual-thresholds approach on sample separation.

**Our method achieves excellent reconstruction of the CLS.** The objective of CLS reconstruction is to ensure the ground-truth label is retained within the CLS while minimizing the size of the CLS. We conducted experiments on CIFAR10 ($\eta = 0.5$, $\gamma = 0.3$) to validate this in Fig. 3(b), where we can find the reconstructed CLS size decreases while ensuring a high ground-truth label hit rate with the progression of training iterations. The above observation validates the effectiveness of our CLS reconstruction method.

**Our method narrows generalization error bound.** Our approach is motivated by the generalization error bound in Theorem 1. From Theorem 1, the generalization error bound of NPLL is positively correlated with factor $1 - \frac{1-\epsilon}{\alpha}$. Fig. 3(c) indicates the factor $1 - \frac{1-\epsilon}{\alpha}$ is continuously decreasing as the model training progresses which confirms the proposed approach can narrow the generalization error bound.

Table 5: Ablation study of our method (%).

| Method | CIFAR10 $\eta = 0.5$ $\gamma = 0.3$ | CIFAR100 $\eta = 0.05$ $\gamma = 0.3$ |
|---|---|---|
| Ours | **95.41** | **79.88** |
| Ours *v1* | 94.09 | 78.68 |
| Ours *v2* | 95.39 | 75.16 |
| Ours *v3* | 81.32 | 74.62 |
| Papi | 61.98 | 62.72 |

**All the components contribute to the proposed model.** We conducted experiments to assess the effectiveness of each component. Specifically, we tested Ours *v1* by discarding the selected uncertainty samples in model training; Ours *v2* by separating normal samples and noisy samples in an one-off manner, i.e., by maintaining $l = u$ throughout the training process; Ours *v3* by removing CLS reconstruction and using non-CLS as CLS for noisy samples. From Table 5, we can see that all the ingredients of our method contribute to the performance improvement. Comparing Ours *v2* with Ours, it is evident that the two-thresholds separation strategy is important for more challenging datasets, as the performance decreases 4.72% on

CIFAR100. Comparing Ours *v3* with Ours on CIFAR-10 and CIFAR-100 datasets, the performance decreased by 14.09% and 5.26% respectively, indicating that CLS reconstruction is significantly effective in NPLL task.

# 6 RELATED WORK

## 6.1 PARTIAL LABEL LEARNING (PLL)

In PLL, an instance is associated with a candidate label set in which the ground-truth label is concealed. PLL aims to learn a multi-class classifier from the ambiguous candidate label set (CLS) (Hüllermeier & Beringer, 2006; Nguyen & Caruana, 2008). The key to PLL is disambiguation, i.e., finding the correct label from the CLS. The average-based PLL methods (Hüllermeier & Beringer, 2006; Cour et al., 2011) treat all the candidate labels as ground-truth labels equally, while the identification based PLL methods (Zhang et al., 2022; Jia et al., 2023b) treat the ground-truth label as latent variable and try to find it from the candidate labels. Graph-based PLL methods (Wang et al., 2022a; Jia et al., 2023b) use the similarity (or dissimilarity) relationship of features and candidate labels to achieve label disambiguation. Recently, deep learning-based PLL methods (Feng et al., 2020; Wen et al., 2021) have become popular due to their excellent classification performance. For instance, (Lv et al., 2020; Feng et al., 2020; Wen et al., 2021; Zhang et al., 2022) perform disambiguation by dynamically adjusting the confidence of the candidate labels as pseudo-labels to guide model training. PiCO (Wang et al., 2022b) introduces contrastive learning into PLL and disambiguates CLS through class prototypes. CRDPLL (Wu et al., 2022) performs disambiguation on CLS with a consistency regularization. PaPi (Xia et al., 2023) adopts the model outputs for disambiguation and utilizes the results of disambiguation to guide the learning of representations. However, the above PLL methods all assume that the ground-truth label must locate in the candidate set, which may not hold in real-world scenarios as the non-expert annotators may make erroneous judgments.

## 6.2 NOISY PARTIAL LABEL LEARNING (NPLL)

Recently, some researchers have turned their attention to a more practical setting called Noisy Partial Label Learning (Lv et al., 2024) which allows the CLS of some instances do not contain the ground-truth label. The problem is also referred to as Unreliable Partial label learning (UPLL) (Shi et al., 2023; Lv et al., 2024) in some researches. UPLLRS (Shi et al., 2023) proposes a recursive separation strategy to first regard the noisy sample as unlabeled data. Then UPLLRS employs a semi-supervised learning approach to train the model for these unlabeled data, while other PLL samples are tackled by PLL loss. PiCO+ (Wang et al., 2024) detects the noisy samples with KNN and then employs semi-supervised methods to handle these noisy samples. FREDIS (Qiao et al., 2023) not only moves labels from non-CLS to CLS for handling label noise, but also performs disambiguation by moving incorrect labels from CLS to non-CLS. ALIM (Xu et al., 2023) trades off the initial candidate set and model prediction with an adjusting label importance mechanism. However, these methods are designed based on empirical experience without theoretical support. Through our theoretical analysis in Theorem 1, we are able to identify the shortcomings of the above empirically designed methods. Moreover, PiCO+, UPLLRS and FREDIS do not take into account sample uncertainty in separating noisy samples, which easily cause separation errors leading to large noise rate. ALIM incorporates the initial candidate set with model outputs for eliminating label noise, which expands the size of the CLS.

# 7 CONCLUSION

In this work, we have presented a novel method to solve the NPLL problem. Specifically, we theoretically prove that a smaller noise rate and a shorter length of the average candidate label set will reduce the generalization error bound constructed under the NPLL paradigm. To optimize these two factors, we propose a sample separation strategy to segregate highly reliable normal samples, highly reliable noisy samples, and uncertain samples. Based on the sample separation, we perform distinct partial label set reconstructions for these three kinds of samples to reduce the average size of the candidate label set. Our model can act as a plug-in to promote different PLL methods in NPLL, and extensive experiments suggest the salient performance advantage of our method. Furthermore, the ablation study indicates that all the components contribute to the proposed method.

## 8 ACKNOWLEDGMENTS

This work was supported by the National Natural Science Foundation of China under Grants U24A20322, 62225602, and 62176160, in part by the Guangdong Basic and Applied Basic Research Foundation (Grant 2024B1515020109), and by the Big Data Computing Center of Southeast University.

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

# A    Appendix

## A.1    Proof of Theorem 1

**Theorem.** *Assume the loss function $\mathcal{L}(f(\boldsymbol{x}), y)$ is $\rho$-Lipschitz with respect to $f(\boldsymbol{x})$ for all $y \in \mathcal{Y}$ and uppper-bounded by $M$. For noise rate $0 < \epsilon < 1$ and mean CLS size for normal samples $1 < \alpha < C$, for any $\delta > 0$, with probability at least $1 - \delta$, we have*

$$R(\hat{f}) - R(f^*) \leq 2(1 - \frac{1-\epsilon}{\alpha})M + 4\sqrt{2}\rho \sum_{y=1}^{C} \mathfrak{R}_n(\mathcal{H}_y) + 2M\sqrt{\frac{log\frac{2}{\delta}}{2n}}.$$

*Proof.* Before proving the theorem, we first provide two useful lemmas as follows.

We primarily derive the uniform deviation bound between $R(f)$ and $\widehat{R}(f)$, which is simple extension of result in muti-class setting (Xie et al., 2023).

**Lemma 1.** *Suppose that the loss function $\mathcal{L}(f(\boldsymbol{x}), y)$ is $\rho$-Lipschitz with respect to $f(\boldsymbol{x})$ for all $y \in \mathcal{Y}$ and uppper-bounded by $M$. For any $\delta > 0$, with probability at least $1 - \delta$, we have*

$$\left| R(f) - \widehat{R}(f) \right| \leq \sqrt{2}\rho \sum_{y=1}^{k} \mathfrak{R}_n(\mathcal{H}_y) + M\sqrt{\frac{\log\frac{2}{\delta}}{2n}}, \tag{12}$$

where the function space $\mathcal{H}_y$ for the label $y \in \mathcal{Y}$ is $\{h : \boldsymbol{x} \mapsto f_y(\boldsymbol{x}) | f \in \mathcal{F}\}$.

*Proof.* We first define the Rademacher complexity of $\mathcal{L}$ and $\mathcal{F}$ with $n$ training instances as follows:

$$\mathfrak{R}_n(\mathcal{L} \circ \mathcal{F})$$

$$= \mathbb{E}_{\boldsymbol{x}, \boldsymbol{y}, \boldsymbol{\sigma}} \left[ \sup_{f \in \mathcal{F}} \sum_{i=1}^{n} \sigma_i(\mathcal{L}(f(\boldsymbol{x}_i), y_i)) \right]$$

$$\leq \sqrt{2}\rho \sum_{y=1}^{C} \mathfrak{R}_n(\mathcal{H}_y). \tag{13}$$

We apply the Rademacher vector contraction inequality (Maurer, 2016) in the second line.

Then, we proceed the proof by showing that the one direction $\sup_{f \in \mathcal{F}} R(f) - \widehat{R}(f)$ is bounded with probability at least $1 - \delta/2$, and the other direction can be proved similarly. Note that replacing an example $(x_i, y_i)$ leads to a change of $\sup_{f \in \mathcal{F}} R(f) - \widehat{R}(f)$ at most $\frac{M}{n}$ due to the $\mathcal{L}$ is bounded by $M$. According to *McDiarmid's inequality* (McDiarmid, 1989), for any $\delta > 0$, with probability at least $1 - \delta/2$, we have

$$\sup_{f \in \mathcal{F}} R(f) - \widehat{R}(f) \leq \mathbb{E} \left[ \sup_{f \in \mathcal{F}} R(f) - \widehat{R}(f) \right] + M\sqrt{\frac{\log\frac{2}{\delta}}{2n}}. \tag{14}$$

According to the result in (Maurer, 2016) that shows $\mathbb{E}[\sup_{f \in \mathcal{F}} R(f) - \widehat{R}(f)] \leq 2\mathfrak{R}_n(\mathcal{F})$, and further taking the other direction $\sup_{f \in \mathcal{F}} \widehat{R}(f) - R(f)$ into account. With probability at least $1 - \delta$, we have

$$\left| R(f) - \widehat{R}(f) \right| \leq 2\sqrt{2}\rho \sum_{y=1}^{k} \mathfrak{R}_n(\mathcal{H}_y) + M\sqrt{\frac{\log\frac{2}{\delta}}{2n}}, \tag{15}$$

which completes the proof.

**Lemma 2.** *Assume the loss function $\mathcal{L}(f(\boldsymbol{x}), y)$ is $\rho$-Lipschitz with respect to $f(\boldsymbol{x})$ for all $y \in \mathcal{Y}$ and uppper-bounded by $M$. For noise rate $0 < \epsilon < 1$ and mean CLS size for normal samples $1 < \alpha < C$, we have*

$$\left| \widehat{R}'(f) - \widehat{R}(f) \right| \leq (1 - \frac{1-\epsilon}{\alpha})M. \tag{16}$$

*Proof.* Let's first expand $\widehat{R}'(f)$:

$$
\begin{aligned}
\widehat{R}'(f) =&\frac{1}{n}\sum_{i=1}^{n}\mathcal{L}(f(\boldsymbol{x}_i),\boldsymbol{y}_i) \\
&+ \frac{1}{n}\sum_{i=1}^{n}\mathbb{I}(\boldsymbol{y}_i \in Y_i)\left[\sum_{c \in Y_i, c \neq \boldsymbol{y}_i}\frac{1}{|Y_i|}\mathcal{L}(f(\boldsymbol{x}_i),c) - \frac{|Y_i|-1}{|Y_i|}\mathcal{L}(f(\boldsymbol{x}_i),\boldsymbol{y}_i)\right] \\
&+ \frac{1}{n}\sum_{i=1}^{n}\mathbb{I}(\boldsymbol{y}_i \notin Y_i)\left[\sum_{c \in Y_i}\frac{1}{|Y_i|}\mathcal{L}(f(\boldsymbol{x}_i),c) - \mathcal{L}(f(\boldsymbol{x}_i),\boldsymbol{y}_i)\right].
\end{aligned}
\tag{17}
$$

Define $m = \sum_{i=1}^{n}\mathbb{I}(\boldsymbol{y}_i \in Y_i)$, which implies the noise rate $\epsilon = \frac{n-m}{n}$. Define the function $f(|Y|) = \frac{|Y|-1}{|Y|}$, which is concave. By Jensen's inequality (Abramovich et al., 2004) and the definition of $\alpha$ ($\alpha = \frac{1}{m}\sum_{j=1}^{m}|Y_j|$), we have

$$
\frac{1}{m}\sum_{j=1}^{m}f(|Y_j|) \leq f\left(\frac{1}{m}\sum_{j=1}^{m}|Y_j|\right) = f(\alpha).
\tag{18}
$$

Then, we prove its lower bound:

$$
\begin{aligned}
\widehat{R}'(f) &\geq \widehat{R}(f) - \frac{1}{n}\sum_{i=1}^{n}\mathbb{I}(\boldsymbol{y}_i \in Y_i)\frac{|Y_i|-1}{|Y_i|}\mathcal{L}(f(\boldsymbol{x}_i),\boldsymbol{y}_i) - \frac{1}{n}\sum_{i=1}^{n}\mathbb{I}(\boldsymbol{y}_i \notin Y_i)\mathcal{L}(f(\boldsymbol{x}_i),\boldsymbol{y}_i) \\
&\geq \widehat{R}(f) - \frac{1}{n}\sum_{i=1}^{n}\mathbb{I}(\boldsymbol{y}_i \in Y_i)\frac{|Y_i|-1}{|Y_i|}M - \frac{1}{n}\sum_{i=1}^{n}\mathbb{I}(\boldsymbol{y}_i \notin Y_i)M \\
&\geq \widehat{R}(f) - \frac{m}{n}\cdot\frac{1}{m}\sum_{j=1}^{m}\frac{|Y_j|-1}{|Y_j|}M - \frac{n-m}{n}M \\
&\geq \widehat{R}(f) - (1-\epsilon)\frac{\alpha-1}{\alpha}M - \epsilon M \\
&\geq \widehat{R}(f) - (1-\frac{1-\epsilon}{\alpha})M,
\end{aligned}
\tag{19}
$$

where the second line holds because $\mathcal{L}(f(\boldsymbol{x}),y)$ is upper-bounded by $M$ ($\mathcal{L}(f(\boldsymbol{x}),y) \leq M$) and the fourth line holds by Eq. (18).

Then, we prove its upper bound:

$$
\begin{aligned}
\widehat{R}'(f) &\leq \widehat{R}(f) + \frac{1}{n}\sum_{i=1}^{n}\mathbb{I}(\boldsymbol{y}_i \in Y_i)\sum_{c \in Y_i, c \neq \boldsymbol{y}_i}\frac{1}{|Y_i|}\mathcal{L}(f(\boldsymbol{x}_i),c) + \frac{1}{n}\sum_{i=1}^{n}\mathbb{I}(\boldsymbol{y}_i \notin Y_i)\sum_{c \in Y_i}\frac{1}{|Y_i|}\mathcal{L}(f(\boldsymbol{x}_i),c) \\
&\leq \widehat{R}(f) + \frac{1}{n}\sum_{i=1}^{n}\mathbb{I}(\boldsymbol{y}_i \in Y_i)\sum_{c \in Y_i, c \neq \boldsymbol{y}_i}\frac{1}{|Y_i|}M + \frac{1}{n}\sum_{i=1}^{n}\mathbb{I}(\boldsymbol{y}_i \notin Y_i)\sum_{c \in Y_i}\frac{1}{|Y_i|}M \\
&\leq \widehat{R}(f) + \frac{1}{n}\sum_{i=1}^{n}\mathbb{I}(\boldsymbol{y}_i \in Y_i)\frac{|Y_i|-1}{|Y_i|}M + \frac{1}{n}\sum_{i=1}^{n}\mathbb{I}(\boldsymbol{y}_i \notin Y_i)M \\
&\leq \widehat{R}(f) + \frac{m}{n}\cdot\frac{1}{m}\sum_{j=1}^{m}\frac{|Y_j|-1}{|Y_j|}M + \frac{n-m}{n}M \\
&\leq \widehat{R}(f) + (1-\epsilon)\frac{\alpha-1}{\alpha}M + \epsilon M \\
&\leq \widehat{R}(f) + (1-\frac{1-\epsilon}{\alpha})M,
\end{aligned}
\tag{20}
$$

where the second line holds because $\mathcal{L}(f(\boldsymbol{x}), y)$ is upper-bounded by $M$ ($\mathcal{L}(f(\boldsymbol{x}), y) \leq M$) the fifth line holds by Eq. (18).

By combining these two sides, we can obtain the following result:

$$\left| \widehat{R}'(f) - \widehat{R}(f) \right| \leq (1 - \frac{1 - \epsilon}{\alpha})M, \tag{21}$$

which concludes the proof.

For any $\delta > 0$, with probability at least $1 - \delta$, we have:

$$
\begin{aligned}
R(\hat{f}) &\leq \widehat{R}(\hat{f}) + 2\sqrt{2}\rho \sum_{y=1}^{C} \Re_n(\mathcal{H}_y) + M\sqrt{\frac{\log \frac{2}{\delta}}{2n}} \\
&\leq \widehat{R}'(\hat{f}) + (1 - \frac{1 - \epsilon}{\alpha})M + 2\sqrt{2}\rho \sum_{y=1}^{C} \Re_n(\mathcal{H}_y) + M\sqrt{\frac{\log \frac{2}{\delta}}{2n}} \\
&\leq \widehat{R}'(f^*) + (1 - \frac{1 - \epsilon}{\alpha})M + 2\sqrt{2}\rho \sum_{y=1}^{C} \Re_n(\mathcal{H}_y) + M\sqrt{\frac{\log \frac{2}{\delta}}{2n}} \\
&\leq \widehat{R}(f^*) + 2(1 - \frac{1 - \epsilon}{\alpha})M + 2\sqrt{2}\rho \sum_{y=1}^{C} \Re_n(\mathcal{H}_y) + M\sqrt{\frac{\log \frac{2}{\delta}}{2n}} \\
&\leq R(f^*) + 2(1 - \frac{1 - \epsilon}{\alpha})M + 4\sqrt{2}\rho \sum_{y=1}^{C} \Re_n(\mathcal{H}_y) + 2M\sqrt{\frac{\log \frac{2}{\delta}}{2n}},
\end{aligned}
\tag{22}
$$

where the first and fifth lines are based on Lemma 1, and second and fourth lines holds due to Lemma 2. The third line hold by the definition of $\hat{f}$. At this point, we have proven the Theorem 1.

### A.2 Solution of Equation 6

$$
\begin{aligned}
\min_{\widehat{Y}} \quad & \sum_{i=1}^{n} I(v_i \neq 0) \left( |\widehat{Y}_i| - \beta \langle \boldsymbol{q_i}, \boldsymbol{S}(\widehat{Y}_i) \rangle \right), \\
\text{s.t.} \quad & \forall i, \text{if } v_i = 1, \widehat{Y}_i \neq \varnothing, \widehat{Y}_i \subseteq Y_i, \\
& \forall i, \text{if } v_i = -1, \widehat{Y}_i \neq \varnothing, \widehat{Y}_i \subseteq \mathcal{Y} - Y_i,
\end{aligned}
$$

We denote $S_{ij}$ the $j$-th element of vector $\boldsymbol{S}(\widehat{Y}_i)$. Then the original optimization objective can be writed as

$$
\begin{aligned}
& \sum_{i=1}^{n} I(v_i \neq 0) \sum_{j=1}^{C} S_{ij} - \beta \sum_{i=1}^{n} I(v_i \neq 0) \sum_{j=1}^{C} q_{ij} S_{ij} \\
= & \sum_{i=1}^{n} I(v_i \neq 0) \sum_{j=1}^{C} (1 - \beta q_{ij}) S_{ij}.
\end{aligned}
$$

Considering $S_{ij} \in \{0, 1\}$, we can obtain that $S_{ij} = 1$ if $q_{ij} > \frac{1}{\beta}$ and $S_{ij} = 0$ otherwise. The inclusion constraint: the ground-truth label of the normal sample ($v_i = 1$) is in the CLS ($\widehat{Y}_i \neq \varnothing, \widehat{Y}_i \subseteq Y_i$), while the ground-truth label of the noisy sample ($v_i = -1$) is in the non-CLS ($\widehat{Y}_i \neq \varnothing, \widehat{Y}_i \subseteq \mathcal{Y} - Y_i$). Therefore we can get the solution that

$$
\widehat{Y}_i = \{j | j \in Y_i \text{ and } q_{ij} > \frac{1}{\beta}\}, \quad \text{if } v_i = 1,
$$

$$
\widehat{Y}_i = \{j | j \in \mathcal{Y} - Y_i \text{ and } q_{ij} > \frac{1}{\beta}\}, \quad \text{if } v_i = -1.
$$

## A.3 Implementation Details

There are four hyper-parameters in our method: $e_w$, $e_{end}$, $\lambda$ and $\beta$. Since our approach is a plug-in method, the number of warm-up rounds $e_w$ is related to the training rounds of the original method. For the CIFAR-10 dataset, we used 0.25 times the number of training rounds of the original method, and used 0.1 times for Treeversity and Benthic ,while for CIFAR-100 and other fine-grained datasets, we used 0.4 times. The uncentainty-end epoch $e_{end}$ equals 0.6 times the number of training rounds of original method. We selected $\lambda$ from $\{0.2, 0.3, 0.4, 0.5\}$ and $\beta$ from $\{1.5, 2, 2.5, 3.0\}$. We used the softmax function as the $Normalize(\cdot)$. Following the standard experimental setup (Xu et al., 2023), we splited a clean validation set from the training set to determine hyper-parameters. Then, we transformed the validation set back to its NPLL form and incorporated them into the training set to optimize the model. Following previous research (He et al., 2024), we employed BLIP-2 (Li et al., 2023b) as the feature extractor based on the open-source library LAVIS (Li et al., 2023a), thereby securing a more reliable $K$-neighbor relationship. For KNN searching, the number of chosen neighbors was set to 5 for all experiments and we employed Faiss (Johnson et al., 2019), a library for efficient similarity search and clustering of dense vectors. For all the methods, we employed the same backbone. For the CIFAR dataset, we utilized ResNet18, while for the fine-grained datasets CUB200 and Flower and the real-world datasets Treeversity and Benthic, we employed ResNet34 and loaded the pre-trained weights from ImageNet for the feature extractor to enhance training efficiency. For all methods, the SGD was used as the optimizer with momentum of 0.9 and weight decay of 0.001. We set the initial learning rate to 0.01 and adjusted it using the cosine scheduler. Due to our approach is a plug-in method, we default to combining our method with PaPi (Xia et al., 2023) for NPLL unless specifically stated. All experiments were implemented with PyTorch (Paszke et al., 2019) and carried out with 6 NVIDIA RTX 3090 GPUs and 8 NVIDIA RTX 4090 GPUs.

## A.4 The Framework of our method

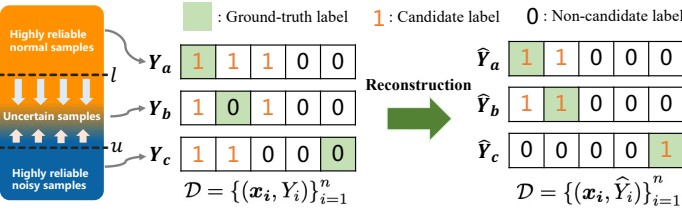

Figure 4: The framework of our method. We employed a dual-threshold approach ($l$ and $u$) to separate the samples into three parts, based on that we proposed CLS reconstruction for faithful shorter CLS for each instance.

## A.5 Other results on CIFAR100

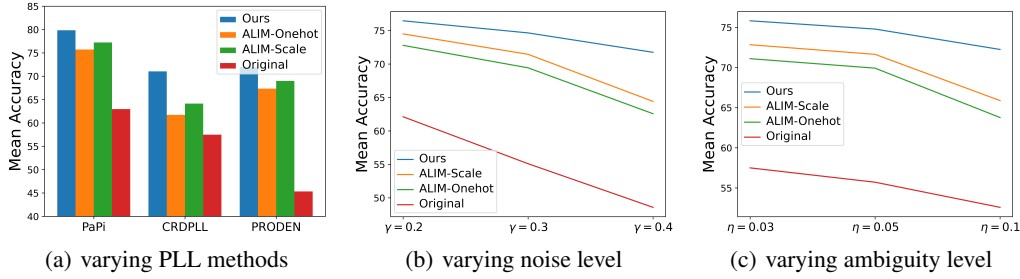

(a) varying PLL methods     (b) varying noise level     (c) varying ambiguity level

Figure 5: The mean accuracy on CIFAR100 of our method and the ALIM across different PLL method, ambiguity levels, and noise levels. The results are derived from the statistics presented in Table 2.

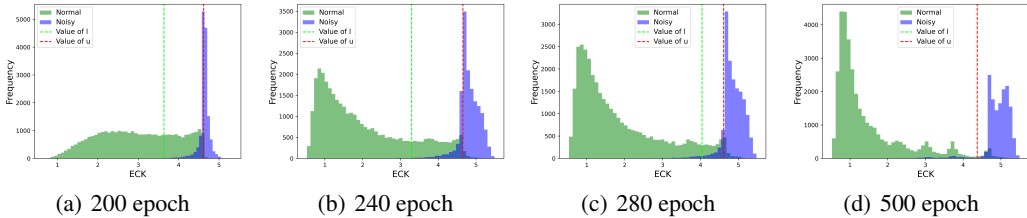

| (a) 200 epoch | (b) 240 epoch | (c) 280 epoch | (d) 500 epoch |

Figure 6: Distribution of the ECK for real normal and noisy samples with increasing training iterations. The experiment is conducted on CIFAR100 ($\eta = 0.05$, $\gamma = 0.3$).

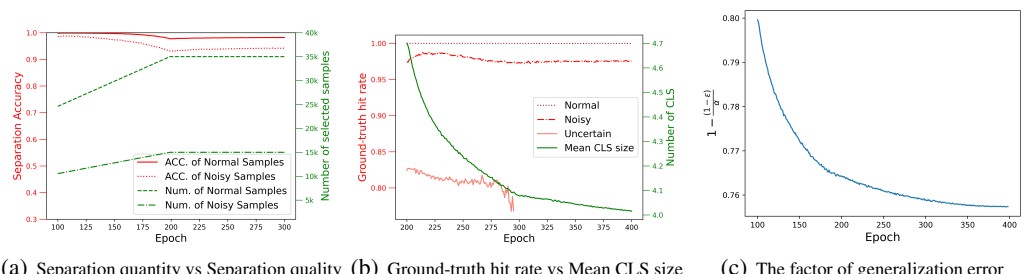

(a) Separation quantity vs Separation quality  (b) Ground-truth hit rate vs Mean CLS size  (c) The factor of generalization error

Figure 7: The experiment is conducted on CIFAR100 ($\eta = 0.05$, $\gamma = 0.3$). (a) demonstrates that with the progression of training iterations, the selection quantity of both normal and noisy samples increases, while the quality is able to be sustained at a high level. (b) reflects that, as training iterations continue, the reconstructed CLS size consistently decreases while ensuring a high ground-truth label hit rate. There are no uncertain samples after 200 epoch, i.e., $e_{end} = 300$. (c) indicates that, as the model training progresses, the generalization error bound is continuously decreasing, with $1 - \frac{1-\epsilon}{\alpha}$ being an important factor derived from Section 3 for the generalization error bound.

## A.6 Real-world datasets

We further evaluate our method on two the real-world crowdsourced datasets Treeversity and Benthic. In these datasets, each image is annotated by multiple individuals. For our study, we randomly selected 2 annotations to serve as a sample's CLS to construct a dataset Treeversity2, and randomly selected 3 annotations to construct a dataset Treeversity3. Given the limited number of annotations per sample in the Benthic dataset, we randomly selected 1 annotation for 60% of the samples and 2 annotations for the remaining samples, resulting in the Benthic2# dataset. Below is the detailed information about these datasets:

Table 6: The detailed information of real-world datasets.

| Real-world Datasets | Noise rate | Avg. of CLS size | Sample num | Classes | Input size |
|---|---|---|---|---|---|
| Treeversity3 | 0.07 | 1.72 | 9826 | 6 | 224 * 224 |
| Treeversity2 | 0.14 | 1.45 | 9826 | 6 | 224 * 224 |
| Benthic2# | 0.12 | 1.12 | 4867 | 8 | 112 * 112 |

We used the ResNet34 as the backbone for all methods and loaded the pre-trained weights from ImageNet to enhance training efficiency. The experimental results in the Table 4 indicate that our method still significantly outperforms the current SOTA methods on real-world datasets. Especially, in more challenging scenarios like Treeversity2, the performance gap is even more significant.

## A.7 Separation accuracy comparisons to current NPLL methods

The $\gamma$ is noise level and $\eta$ is the ambiguity level in Table 7. Our method outperforms other NPLL methods with a substantial gap especially when noise level is high, which validates the superiority

Table 7: The separation accuracy of **normal/noisy samples** for each NPLL method.

| Method\Dataset | CIFAR10 ($\gamma$=0.2, $\eta$=0.5) | CIFAR10 ($\gamma$=0.3, $\eta$=0.5) | CIFAR10 ($\gamma$=0.4, $\eta$=0.5) |
|---|---|---|---|
| PiCO+ | 99.21%/48.90% | 96.25%/69.14% | 83.92%/76.11% |
| ALIM-Onehot | 98.53%/94.44% | 97.31%/93.36% | 86.80%/79.33% |
| Ours | **99.35%/97.85%** | **98.91%/97.08%** | **92.68%/88.05%** |

of ECK on sample separation and demonstrates that our method is more robust against label noise compared to other NPLL methods.

## A.8 LIMITATION

While our method has achieved superior performances in handling NPLL tasks, it still has shortcoming. Due to our method being applied during the training process of a given PLL model, it necessitates additional computations for sample separation and CLS reconstruction which incurs extra computational and space costs. However, correspondingly, as our method gradually reduces the length of the CLS during reconstruction, it can accelerate the disambiguation process of the PLL methods, thereby decreasing the number of training iterations. This can mitigate the issue of increased training time per iteration to a certain extent.

## A.9 PARAMETER SENSITIVITY ANALYSIS

In this section, we perform parameter sensitivity analysis on four hyper-parameters: the warm-up epoch ($e_w$), the rate of percentile change between $r_l$ and $r_u$ ($\lambda$), the trade-off of two objectives on CLS Reconstruction ($\beta$), and the KNN parameter ($K$). Figure 8 presents the results of the hyperparameters sensitivity study conducted on various hyper-parameters on CIFAR10 ($\eta = 0.5$, $\gamma = 0.3$) and CIFAR100 ($\eta = 0.05$, $\gamma = 0.3$). To investigate the influence of noise on hyper-parameter sensitivity, we further conducted sensitivity analyses at higher noise level on CIFAR100 ($\eta = 0.05$, $\gamma = 0.4$) on Figure 9. The results show that $\beta$ has stable performance across a range of values, with CIFAR10 ($\eta = 0.5$, $\gamma = 0.3$) achieving the best accuracy (95.46%) at $\beta = 1.5$, and CIFAR100 ($\eta = 0.05$, $\gamma = 0.3$) peaking at 80.39% when $\beta = 2.0$. The KNN parameter $K$ shows stable results with CIFAR10 ($\eta = 0.5$, $\gamma = 0.3$) achieving 95.55% at $K = 7$ and CIFAR100 ($\eta = 0.05$, $\gamma = 0.3$) reaching 80.11% at $K = 5$. The comparison between Figures 8 and 9 reveals that the hyperparameters $\beta$ and $K$ demonstrate robustness against noise. The model achieves optimal performance when $\beta$ is set between 1.5 and 2 and $K$ is set between 5 and 7 across different noise levels. For $\lambda$, CIFAR10 ($\eta = 0.5$, $\gamma = 0.3$) and CIFAR100 ($\eta = 0.05$, $\gamma = 0.3$) reach their highest accuracy (95.13% and 80.46%) at $\lambda = 0.4$, with performance declining at lower or higher values. Its selection is influenced by the noise rate in the dataset, as $\lambda$ controls the percentile change rate between normal and noisy samples during the separation process. Therefore, we recommend increasing $\lambda$ under high noise conditions and reducing it under low noise scenarios. The warm-up epoch $e_w$ is crucial for training stability. The insufficient pretraining with a small $e_w$ may lead to poor discrimination, hindering the generation of reliable pseudo-labels. Conversely, an excessively large $e_w$ can result in overfitting to noisy samples, reducing the model's ability to correct errors in later stages. Nonetheless, setting $e_w$ within the range of 100 to 200 consistently yields satisfactory performance across different datasets and noise levels. For instance, on CIFAR10 ($\eta = 0.5$, $\gamma = 0.3$), the best accuracy (96.23%) is achieved at $e_w = 150$, while CIFAR100 ($\eta = 0.05$, $\gamma = 0.3$) peaks at 80.19% with $e_w = 150$. Overall, while $\beta$ and $K$ are robust, tuning of $\lambda$ and $e_w$ is necessary.

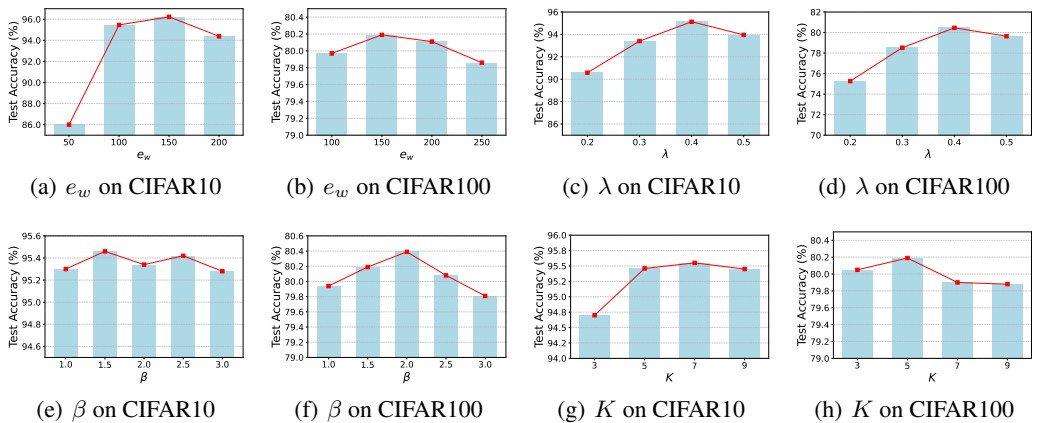

Figure 8: Parameter sensitivity analysis on CIFAR10 ($\eta = 0.5$, $\gamma = 0.3$) and CIFAR100 ($\eta = 0.05$, $\gamma = 0.3$).

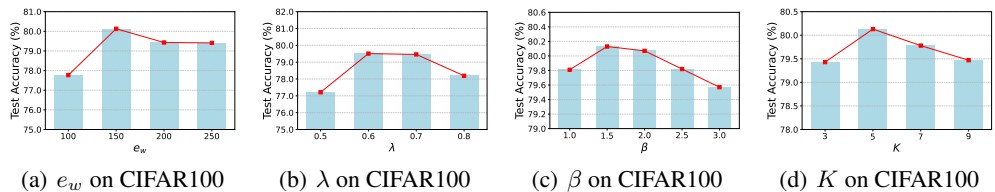

Figure 9: Parameter sensitivity analysis on CIFAR100 ($\eta = 0.05$, $\gamma = 0.4$).

