# OpenReview forum: "Noise Separation guided Candidate Label Reconstruction for Noisy Partial Label Learning"
_ICLR.cc/2025/Conference — ICLR 2025 Poster_

### Official Review · Reviewer_RE2k · 2024-11-01

**Soundness:** 3
**Presentation:** 3
**Contribution:** 3
**Rating:** 6
**Confidence:** 2

**Summary:**

The paper introduces a new framework for noisy partial label learning (NPLL) that combines progressive sample separation and candidate label reconstruction to reduce noise rates and improve classification accuracy, achieving a good  performance enhancement on the CIFAR10 dataset compared to state-of-the-art methods.

**Strengths:**

1. This paper is clearly written and easy to understand.

2. The paper provides the generalization error bound for the classifier constructed under NPLL.

3. Extensive experimental results demonstrate that our method significantly outperforms the current state-of-the-art (SOTA) methods.

**Weaknesses:**

1. For ablation study,  $v_2$ seems ineffective in separating normal samples and noisy samples in CIFAR10.

2. For Eq.(3), the author calculates the ECK of each sample $x_i$ by CE and uses $\widetilde q$ as target. What about using a KNN-based pseudo-label as the target instead?

3. The sentence below Eq.(10), $c^\prime$ can be  $c^\prime = \mathrm{arg min}_{j \in Y_i}$ instead of   arg max

4. The datasets adopt in the experiment are few and small. It would be more effective to utilize larger datasets to further validate the effectiveness of the proposed method.

**Questions:**

Please refer to the weakness

---

> ### Author Response · Authors · 2024-11-21
>
> **W1:**
>
> Due to the relative simplicity of the CIFAR10 dataset on setting $\eta$=0.5, $\gamma$=0.3 and the benefit of our proposed ECK metric, a well-pretrained single-threshold strategy can still achieve satisfactory performance. As shown in the table below, the gap between *Ours* and *Ours v2* widens as the situation becomes more challenging, such as with more difficult dataset (CIFAR100) or higher noise rates. This demonstrates the effectiveness of our dual-threshold strategy, particularly in handling more complex scenarios.
>
> | Method \ Dataset | CIFAR10 ( $\eta$=0.5, $\gamma$=0.3) | CIFAR10 ( $\eta$=0.5, $\gamma$=0.4) | CIFAR100 ( $\eta$=0.05, $\gamma$=0.3) | CIFAR100 ( $\eta$=0.05, $\gamma$=0.4) |
> | ---------------- | ----------------------------------- | ----------------------------------- | ------------------------------------- | ------------------------------------- |
> | *Ours*           | 95.41%                              | 81.85%                              | 78.23%                                | 78.10%                                |
> | *Ours V2*        | 95.39%                              | 79.34%                              | 75.16%                                | 73.91%                                |
>
>
>
> **W2:**
>
> ECK is used to measure the inconsistency error between KNN-based pseudo-labels and CLS-based pseudo-labels. It serves as a tool to distinguish normal samples from noisy samples. As you suggested, we conducted experiments in which the roles of KNN-based pseudo-labels and CLS-based pseudo-labels were interchanged during the calculation of ECK. The results indicate that Reverse ECK outperforms the current SOTA method ALIM, highlighting the effectiveness of our proposed approach. However, the original method achieves better performance than the reverse one.
>
> | Method\Dataset | CIFAR10 ( $\eta$=0.5, $\gamma$=0.3) | CIFAR100 ( $\eta$=0.05, $\gamma$=0.3) |
> | -------------- | ----------------------------------- | ------------------------------------- |
> | Original ECK   | 95.41%                              | 79.88%                                |
> | Reverse ECK    | 94.73%                              | 77.77%                                |
> | ALIM           | 92.26%                              | 75.67%                                |
>
>
>
> **W3:**
>
> We appreciate your observation regarding this typo. It has been corrected in the latest version of the paper, and we will conduct a thorough review to avoid the typos in the manuscript.
>
>
>
> **W4:**
>
> Thank you for your suggestions. We conducted extensive experiments across a wide range of datasets. Our method was first evaluated on two benchmark datasets, CIFAR10 and CIFAR100, with various noise and ambiguity levels. Additionally, we tested on three fine-grained datasets (CIFAR100H, CUB200, Flower) and validated our approach on two real-world crowdsourced datasets (Treeversity and Benthic). Therefore, we believe our experimental evaluation is both comprehensive and diverse.
>
> However, as you suggest, we further conduct experiments on a large WebVison dataset which contains ∼2.4 million images based on queries generated from the 1,000 ImageNet classes. As this dataset was crawled from the web, its training set inherently contains noise. Thus, generating the NPLL dataset requires only the specification of the noise (ambiguity) rate. It is important to note that the WebVision test set is noise-free. To expedite training, we utilized samples from the first 50 classes for both training and testing. The experimental results of test accuracies of each method presented in the table below further validate the effectiveness of our proposed method on larger dataset.
>
> | Dataset \ Method      | Ours       | ALIM-Onehot | ALIM-Scale | PiCO+  |
> | --------------------- | ---------- | ----------- | ---------- | ------ |
> | WebVison ($\eta$=0.1) | **73.88%** | 70.72%      | 71.44%     | 71.20% |

---

> ### Author Response · Authors · 2024-11-25
>
> Dear reviewer RE2k,
>
> Thanks again for your time and efforts in reviewing this paper and the valuable comments on improving its quality. As the reviewer-author discussion deadline approaches, please take a few minutes to read the rebuttal. If you have further concerns, we are happy to provide more explanations. Thanks.
>
> Regards from the authors.

---

> ### Comment · Reviewer_RE2k · 2024-11-26
> **Reply Official Comment by Authors**
>
> Thanks for the authors' response.
> For Q2, I am not sure whether the authors misunderstand my quesiton.  I mean replace the $p_i$ in Eq.(2) with the $q_i$ computed by Eq.(1).
>
> Moreover, there are two pseudo labels: CLS-based pseudo label and KNN-based pseudo-label, which of them can be the better one in your setting.

---

> > ### Author Response · Authors · 2024-11-27
> >
> > (1)
> >
> > In response to your suggestion, we have supplemented the experiment **Ours V0**, in which $p_i$ in Eq. (2) is replaced by $q_i$ (KNN-based pseudo-label) computed according to Eq. (1). As shown in the table below, **Ours V0** also achieves promising results; however, our original method demonstrates better performance.
> >
> > | Dataset \ Method                      | Ours original | Ours V0 |
> > | ------------------------------------- | ------------- | ------- |
> > | cifar10  ($\eta$=0.5 $\gamma$ =0.3)   | 95.41%        | 95.19%  |
> > | cifar100  ($\eta$=0.05 $\gamma$ =0.3) | 79.88%        | 79.84%  |
> >
> >
> >
> > (2)
> >
> > The accuracy of the CLS pseudo-label is usually lower than the KNN-based pseudo labels especially in the high noisy rate situations. Specifically, when the sample is a noisy sample (i.e., the ground-truth label is not in the CLS), the CLS pseudo-label is completely incorrect. In contrast, KNN pseudo-labels are constructed based on feature space that not directly affected by the noise CLS. Fig. 4(b) of the paper proves that the reconstructed CLS based on the KNN-pseudo labels $\boldsymbol{q_i}$ in Eq. (6) maintains an almost 100% ground-truth hit rate during model training, suggesting the rationality of the current approach (using KNN-based pseudo labels in CLS reconstruction).
> >
> > To further support my explanation, we present the accuracy of CLS pseudo-labels and KNN-based pseudo-labels on the training set across different datasets and noise levels in the tables below. We observe that KNN-based pseudo-labels outperform CLS pseudo-labels, particularly as the noise level increases, with the performance gap between the two becoming more significant.
> >
> > | Dataset \ pseudo label              | KNN-based | CLS-based |
> > | ----------------------------------- | --------- | --------- |
> > | cifar10  ($\eta$=0.5 $\gamma$ =0.2) | 98.83%    | 79.28%    |
> > | cifar10  ($\eta$=0.5 $\gamma$ =0.3) | 97.97%    | 69.53%    |
> >
> > | Dataset \ pseudo label                | KNN-based | CLS-based |
> > | ------------------------------------- | --------- | --------- |
> > | cifar100  ($\eta$=0.05 $\gamma$ =0.2) | 94.02%    | 78.69%    |
> > | cifar100  ($\eta$=0.05 $\gamma$ =0.3) | 92.06%    | 69.21%    |
> >
> > **Although the accuracy of CLS-based pseudo-label is not high, our objective in introducing CLS-based pseudo-label is to examine whether the sample is a noisy sample by comparing it with KNN-based pseudo-label.** For noisy samples, there will be a significant difference between CLS pseudo-label and KNN pseudo-label, resulting in a high ECK value. Conversely, if the ECK value is low, it indicates that it is a normal sample. As illustrated in Fig. 3 of the paper, the ECK values for noisy samples are significantly higher than those for normal samples.

---

> > > ### Comment · Reviewer_RE2k · 2024-11-29
> > > **Official Comments by Reviewer RE2k**
> > >
> > > Thanks for the authors' further reply.
> > >
> > > For (1), there is no difference in performance between these two methods. It is better to provide the Standard Deviation.
> > >
> > > Moreover, the authors claim that the KNN-based pseudo labels can be more reliable than CLS-based pseudo-label, however, $\widetilde q_{ij} $ computed by $p_i$ is used as the target in Eq.(3), why?

---

> > > > ### Author Response · Authors · 2024-11-30
> > > >
> > > > **(1) Standard Deviation.**
> > > >
> > > > Following your suggestion, the table below presents the mean and standard deviation of the results (%) from three experiments. As observed, **Ours V0** ($p_i$ in Eq. (2) is replaced by $q_i$ ) achieves competitive results; however, our proposed method demonstrates slightly superior performance with a smaller variance.
> > > >
> > > > | Dataset \ Method                      | Ours original        | Ours V0          |
> > > > | ------------------------------------- | -------------------- | ---------------- |
> > > > | cifar10  ($\eta$=0.5 $\gamma$ =0.3)   | **95.02 $\pm$ 0.42** | 94.64 $\pm$ 0.54 |
> > > > | cifar100  ($\eta$=0.05 $\gamma$ =0.3) | **80.18 $\pm$ 0.34** | 79.68 $\pm$ 0.51 |
> > > >
> > > > The final difference caused by using $p_i$ or $q_i$ in Eq. (2) is minimal because model output will be close to the KNN pseudo label when the model is well-trained. However, the key component of CLS-based pseudo-label construction (Eq. (2)) is the CLS constraint, $S(Y_i)$, which restricts the pseudo-label probability distribution to CLS space. When CLS is inaccurate, it leads to inconsistencies between KNN-based and CLS-based pseudo-labels (a more detailed explanation is provided in the response below (2)). As shown in the table below, we validated our claim through the *Ours discard S(Y)* experiment that we discarded $S(Y_i)$ in Eq. (2), which reveals a significant decline in performance when $S(Y_i)$ is excluded.
> > > >
> > > > | Dataset \ Method                      | Ours original | Ours discard S(Y) |
> > > > | ------------------------------------- | ------------- | ----------------- |
> > > > | cifar10  ($\eta$=0.5 $\gamma$ =0.3)   | **95.41%**    | 73.61%            |
> > > > | cifar100  ($\eta$=0.05 $\gamma$ =0.3) | **79.88%**    | 65.85%            |
> > > >
> > > > **(2) ECK in Eq. (3) is not a loss but a metric to distinguish the normal samples and the noisy samples.**
> > > >
> > > > ECK (Eq. (3)) is a metric that quantifies the distance between CLS-based pseudo-labels and KNN-based pseudo-labels using cross-entropy. Therefore, the core idea of ECK lies not in the accuracy of the two pseudo-labels, but in the difference between them. When a sample is noisy (i.e., its ground-truth label does not belong to the CLS), the probability distribution of the CLS-based pseudo-label is misaligned with an incorrect CLS. In contrast, KNN-based pseudo-labels are derived from the feature space, which is less affected by the noise in the CLS. Consequently, the distributions of the two pseudo-labels diverge significantly, leading to a larger ECK value. For normal samples, however, the distributions of CLS-based and KNN-based pseudo-labels are more consistent, resulting in smaller ECK values. As illustrated in Fig. 3 of the paper, the ECK values for noisy samples are significantly higher than those for normal samples which validates our statement above.
> > > >
> > > > Cross-entropy provides two forms for measuring the distance between probability distributions: $-\widetilde{q}\log q$ (denoted as *Ours Original*) and $-q \log \widetilde{q}$ (denoted as *Ours Reverse*). We selected *Ours Original* because enforced CLS constraints on caculating CLS-based pseudo-labels $\widetilde{q}$ (Eq. (2)) can result in zero probabilities for non-CLS categories, leading to undefined values ($\log 0$). While adding a small constant (e.g. 1e-6) can mitigate this issue, it may introduce instability, as the negative logarithm of such a small constant generates a large value. Experimental results confirm that *Ours Original* achieves the best performance, as shown in the table below. While *Ours Reverse* surpasses the current SOTA method *ALIM*, validating the effectiveness of our strategy, the *Ours Original* demonstrates even superior performance.
> > > >
> > > > | Method\Dataset  | CIFAR10 ( $\eta$=0.5, $\gamma$=0.3) | CIFAR100 ( $\eta$=0.05, $\gamma$=0.3) |
> > > > | --------------- | ----------------------------------- | ------------------------------------- |
> > > > | *Ours Original* | **95.41%**                          | **79.88%**                            |
> > > > | *Ours Reverse*  | 94.73%                              | 77.77%                                |
> > > > | *ALIM*          | 92.26%                              | 75.67%                                |
> > > >
> > > >
> > > >
> > > > Thank you very much for your reply. If you have any further concerns, we would be pleased to provide additional clarification.

---

> > > > > ### Comment · Reviewer_RE2k · 2024-12-02
> > > > > **Official Comments by Reviewer RE2k**
> > > > >
> > > > > Thanks for your response.  I will keep my positive score.

---

### Official Review · Reviewer_Xu7T · 2024-11-01

**Soundness:** 3
**Presentation:** 3
**Contribution:** 2
**Rating:** 5
**Confidence:** 4

**Summary:**

The paper addresses the problem of Noisy Partial Label Learning (NPLL), a challenging subset of weakly supervised learning where instances are annotated with a set of candidate labels, only one of which is correct. The authors theoretically demonstrate that the generalization error of a classifier constructed under the NPLL paradigm is constrained by the noise rate and the average length of the candidate label set. Building on this theoretical foundation, they propose a novel NPLL framework designed to distinguish noisy samples from normal ones, thereby reducing the noise rate and reconstructing shorter candidate label sets for both categories.

**Strengths:**

1. The paper presents a groundbreaking contribution to the field of weakly supervised learning, specifically addressing the Noisy Partial Label Learning (NPLL) problem. The authors’ novel approach to bounding the generalization error of classifiers under NPLL is a significant theoretical advancement. The proposed method, which integrates progressive sample separation and CLS reconstruction, demonstrates a clear and innovative approach to solving NPLL. This originality in problem formulation and theoretical analysis is commendable.
2. The article's structure is well-organized and easy to comprehend.
3. The experiments are generally comprehensive.
4. Theoretical analysis contributes to establishing the effectiveness of the proposed model.

**Weaknesses:**

1.The computational efficiency of the proposed model is a concern.  The requirement to train the classifier on the entire dataset at every epoch introduces a high time complexity, which may limit the practical applicability of the method, especially for large datasets.
2.The computational efficiency of the proposed model is a concern.  The requirement to train the classifier on the entire dataset at every epoch introduces a high time complexity, which may limit the practical applicability of the method, especially for large datasets.
3.When there are many noisy samples, for the calculation of KNN, if the neighboring samples are also noisy, is this method reliable?

**Questions:**

Could the authors provide a more detailed explanation or theoretical justification for the assumption that reliable samples become increasingly reliable, while unreliable samples become increasingly unreliable as training progresses?

---

> ### Author Response · Authors · 2024-11-21
>
> **W1 & W2:**
>
> W1 and W2 are duplicated.
>
> We would like to address a misunderstanding regarding our method. **Our approach does not involve training a new classifier in each epoch but is instead a plug-in strategy designed to be integrated into any existing PLL model for defeating noise.** In each epoch, our method refines the dataset before introducing it to the PLL model. This refinement process requires only the sample features and predicted probability vectors stored from the previous epoch of the PLL model, only resulting in an additional computational complexity for KNN search and ranking. **Besides, the code of our method has been submitted to the Supplementary Material. You could check and run the code to get more implementation details.**
>
> By utilizing the Faiss library for high-speed KNN search on GPUs, the additional computational overhead is minimized. To validate our claim, we integrated our method into Papi (Xia et al., 2023) and conducted experiments on CIFAR100 with an NVIDIA RTX 3090 GPU. The results showed that the time per epoch increases only 8.67%, supporting the efficiency and practicality of our approach. Although the integration of our method introduces a slight computational time overhead, it yields substantial performance improvements. As presented in Table 2, on the CIFAR-100 dataset ($\eta=0.1$), our approach outperforms the original method (PaPi) with accuracy gains of **12.14%**, **16.25%**, and **22.50%** under noise rates of 0.2, 0.3, and 0.4, respectively.
>
>
>
> **W3:**
>
> Higher noise rates intensify the difficulty of the NPLL problems, as evidenced by a significant performance decline across all methods on Table 1 of the manuscript. However, at higher noise levels, the performance gaps between our method and other methods become larger. For example, on CIFAR100 with $\eta=0.1$, our method improves the best compared method by **4.71%**, **8.15%** and **11.57%** on three different noise levels $\gamma=0.2,0.3$ and $0.4$ respectively. The above analysis proves that our method is more reliable than the current methods under more noisy samples.
>
>
>
>
>
> **Q:**
>
> We have never made the claim that "unreliable samples become increasingly unreliable." On the contrary, we believe that as the model undergoes training, its capability improves, leading to a gradual reduction in unreliable samples. This enables more reliable distinctions between normal and noisy samples (See Fig. 3 and Fig. 4(a)).
>
> In the early stages of training, the model lacks sufficient capacity to accurately classify samples as normal or noisy due to its limited training. To address this, our approach adopts a dual-threshold sample separation strategy, which uses two thresholds to categorize samples into high-confidence normal samples, high-confidence noisy samples, and uncertain samples. As the training progresses and model's capability improves, the two thresholds converge gradually, ultimately achieving accurate separation.
>
> Figure 3 of the paper illustrates the ECK distribution for normal and noisy samples during training. It shows that as the number of training epochs increases, the model becomes progressively better at distinguishing between normal and noisy samples. Additionally, the number of high-confidence samples (including both high-confidence normal samples and high-confidence noisy samples) steadily grows throughout the training process, eventually leading to the separation of all samples.

---

> ### Author Response · Authors · 2024-11-25
>
> Dear reviewer Xu7T,
>
> Thanks again for your time and efforts in reviewing this paper and the valuable comments on improving its quality. As the reviewer-author discussion deadline approaches, please take a few minutes to read the rebuttal. If you have further concerns, we are happy to provide more explanations. Thanks.
>
> Regards from the authors.

---

> ### Author Response · Authors · 2024-11-27
>
> Dear reviewer Xu7T,
>
> Thank you once again for your time and effort in reviewing our paper and providing valuable feedback to enhance its quality. With the PDF submission deadline approaching tomorrow, we kindly request your review of our rebuttal. If you have any further concerns, we would be pleased to provide more detailed explanations.
>
> Regards from the authors.

---

> > ### Author Response · Authors · 2024-12-01
> >
> > Dear reviewer Xu7T,
> >
> > We sincerely appreciate your review and the time you have dedicated to our manuscript. The other reviewers have already engaged in comprehensive discussions with us, and their concerns have been effectively resolved. As the discussion phase is nearing its end, we would like to inquire if you have any remaining concerns. We would be happy to provide additional clarifications or explanations as needed.
> >
> > Regards from the authors.

---

> ### Author Response · Authors · 2024-12-02
>
> Dear Reviewer **Xu7T**,
>
> As the Reviewer-Author discussion phase is drawing to a close, we kindly ask you to review our revisions and responses once more and reconsider your rating. All the other reviewers' concerns have been resolved, and they all gave this paper a positive score.
> We eagerly anticipate your feedback. Thank you.
>
> Best regards,
>
> The Authors

---

### Official Review · Reviewer_1LBQ · 2024-11-03

**Soundness:** 3
**Presentation:** 4
**Contribution:** 3
**Rating:** 6
**Confidence:** 4

**Summary:**

The authors provide the generalization error bound of the classifier constructed under NPLL, which depends on the noise rate and the mean size of candidate label sets for normal samples. Empirically, they propose a novel NPLL method which includes two components: progressive sample separation and CLS reconstruction. Their method can serve as a plug-in for existing PLL methods to enhance their performance on NPLL datasets.

**Strengths:**

1. From generalization error bounds in NPLL problem, they find that the lower noise rate and smaller candidate label are two key
factors to solve the NPLL problem.

2. By iterative learning, their method can effectively reduce the noise rate while simultaneously decreasing the size of CLS.

3. Extensive experimental results validate that our method outperforms the current state-of-the-art (SOTA) methods by a large margin

**Weaknesses:**

The authors did not compare with the recent work (Lv et al., 2024).

**Questions:**

1. Can we use the prediction from the network $p_j$ instead of the KNN-based $q_j$ in Equ. (6)? Are there any difference in performance? In other words, can you empirically show that the KNN-based pseudo-label is better than probability prediction vector from the network?
2. Can you compare the performance with (Lv et al., 2024)?

---

> ### Author Response · Authors · 2024-11-21
>
> **W1 & Q2:**
>
> The main contribution of the BABS (Lv et al., 2024) lies in its exploration of the average-based strategy (ABS) in PLL and provide a theoretical robustness analysis for the average PLL losses under ABS. However, BABS have already been outperformed by the method UPLLRS (Shi et al., 2023) that we have outperformed.
>
> But, as suggested, we compare with it in the Table below. In BABS, it was demonstrated that bounded loss functions, such as Mean Absolute Error (MAE) and Mean Square Error (MSE), exhibit robustness to NPLL. Among these loss functions, MAE achieves the best experimental performance in this work. Therefore, MAE is adopted as the baseline in the experiment. The results show our method outperforms  BABS significantly. And the perfermance gap extends under more difficult situations.
>
> | Method\Dataset | CIFAR10 ($\gamma$=0.3, $\eta$=0.3) | CIFAR10 ($\gamma$=0.3, $\eta$=0.5) | CIFAR100 ($\gamma$=0.3, $\eta$=0.05) | CIFAR100 ($\gamma$=0.3, $\eta$=0.1) |
> | -------------- | ---------------------------------- | ---------------------------------- | ------------------------------------ | ----------------------------------- |
> | BABS (MAE)     | 41.61% ± 2.11%                     | 27.88%± 2.58%                      | 21.45% ± 0.74                        | 16.99% ± 2.06%                      |
> | UPLLRS         | 93.85 $\pm$  0.31                  | 91.55 $\pm$ 0.38                   | 70.31 $\pm$ 0.22                     | 68.60 $\pm$ 0.25                    |
> | Ours           | **96.80% $\pm$  0.14%**            | **94.54%  $\pm$ 1.84%**            | **80.17% $\pm$ 0.20%**               | **79.21% $\pm$ 0.88%**              |
>
>
>
> **Q1:**
>
> The accuracy of the CLS pseudo-label is usually lower than the KNN-based pseudo labels especially in the high noisy rate situations. Specifically, when the sample is a noisy sample (i.e., the ground-truth label is not in the CLS), the CLS pseudo-label is completely incorrect. In contrast, KNN pseudo-labels are constructed based on feature space that not directly affected by the noise CLS. Fig. 4(b) of the paper proves that the reconstructed CLS based on the KNN-pseudo labels $\boldsymbol{q_i}$ in Eq. (6) maintains an almost 100% ground-truth hit rate during model training, suggesting the rationality of the current approach.
>
> Besides, as suggested, we supplemented the following experiments: 1. KNN-based: the CLS reconstruction method used in our paper; 2. CLS-based: using CLS-based pseudo-labels for CLS reconstruction; 3. Fusion: combining CLS-based pseudo-labels and KNN-based pseudo-labels for CLS reconstruction. The following experimental results further validate the correctness of our approach.
>
> | Dataset\Method                        | KNN-based (Our) | CLS-based | Fusion |
> | ------------------------------------- | --------------- | --------- | ------ |
> | cifar10  ($\eta$=0.5 $\gamma$ =0.3)   | **95.41%**      | 72.32%    | 87.41% |
> | cifar100  ($\eta$=0.05 $\gamma$ =0.3) | **79.88%**      | 62.33%    | 68.65% |
>
> Although the accuracy of CLS-based pseudo-label is not high, our objective in introducing CLS-based pseudo-label is to examine whether the sample is a noisy sample by comparing it with KNN-based pseudo-label. For noisy samples, there will be a significant difference between CLS pseudo-label and KNN pseudo-label, resulting in a high ECK value. Conversely, if the ECK value is low, it indicates that it is a normal sample. As illustrated in Fig. 3 of the paper, the ECK values for noisy samples are significantly higher than those for normal samples.

---

> ### Author Response · Authors · 2024-11-25
>
> Dear reviewer 1LBQ,
>
> Thanks again for your time and efforts in reviewing this paper and the valuable comments on improving its quality. As the reviewer-author discussion deadline approaches, please take a few minutes to read the rebuttal. If you have further concerns, we are happy to provide more explanations. Thanks.
>
> Regards from the authors.

---

> ### Author Response · Authors · 2024-11-27
>
> Dear reviewer 1LBQ,
>
> Thank you once again for your time and effort in reviewing our paper and providing valuable feedback to enhance its quality. With the PDF submission deadline approaching tomorrow, we kindly request your review of our rebuttal. If you have any further concerns, we would be pleased to provide more detailed explanations.
>
> Regards from the authors.

---

> ### Comment · Reviewer_1LBQ · 2024-11-27
>
> Thanks for the authors' response. I will keep my score.

---

### Official Review · Reviewer_B6bW · 2024-11-03

**Soundness:** 3
**Presentation:** 3
**Contribution:** 3
**Rating:** 6
**Confidence:** 3

**Summary:**

This paper addresses the problem of noisy partial label learning (NPLL), where the true label may not be included in the candidate label set (CLS). The authors provide a generalization bound on the empirical risk and, based on this, propose an NPLL method that progressively separates samples and reconstructs the CLS, reducing the noise rate while shrinking the CLS size. Experiments are conducted on both synthetic and real-world noisy datasets.

**Strengths:**

1. Building on the generalization error bound, the authors propose a method of progressive sample separation and CLS reconstruction, which reduces the noise rate while simultaneously shrinking the CLS size.
2. The performance of the proposed method is promising in the experiments.
3. The writing is very clear and easy to understand.

**Weaknesses:**

**Weaknesses and questions**

1. Theoretical part.
   * "$f_y(x)$" (Line 125) is not defined in the paper. I assume this represents the predicted probability of the $y$-th class for $x$, following the notation in Xie et al. (2023), as cited by the authors in Appendix 1?
   * The authors state, "Theorem 1 shows that the empirical risk minimizer $\hat{f}$ converges to the true risk minimizer $f^*$ as $n \rightarrow \infty$, $\epsilon \rightarrow 0$, and $\alpha \rightarrow 1$" (Line 138). However, even under these conditions, the Rademacher complexity will not necessarily converge to zero, which implies that $R(\hat{f})$ does not necessarily converge to $R(f^)$. Furthermore, even if $R(\hat{f})$ does converge to $R(f^*)$, this does not guarantee that the minimizer $\hat{f}$ itself converges to $f^*$.
2. Proposed Method and Experimental Part
   * Could you clarify the rationale behind the statement, "When the ground-truth label of a sample is included in the candidate labels, the CLS-based pseudo-labels and KNN-based pseudo-labels tend to be similar... however, when the ground-truth label of a sample is in the non-candidate labels, there will be a significant discrepancy between the CLS-based pseudo-label and KNN-based pseudo-label" (Lines 190-195)?
   * In constructing the KNN-based pseudo-label and CLS-based pseudo-label for $x_i$, model probability prediction vectors and features are used. How does the warm-up stage affect this process? What if the warm up stage that relies on noisy labels is not good enough to produce meaningful features, especially when the noise ratio is high?

**Questions:**

See above.

---

> ### Author Response · Authors · 2024-11-21
>
> **Q1(a):**
>
> Thank you for highlighting the undefined notation $f _ y(x)$ in Line 125. You are correct that this term is intended to represent the predicted probability of the $y$-th class for $x$. We have explicitly defined  $f _ y(x)$ upon its first appearance in the updated paper (PDF). This clarification will ensure that readers can understand the notation without ambiguity.
>
>
>
> **Q1(b):**
>
> Yes, as pointed out by you, the Rademacher complexity does not necessarily converge to zero. Theorem 1 aims to demonstrate that as the sample size increases, along with a reduction in noise levels and mean CLS size, the gap between the generalization error $R(\hat{f})$  of the model $\hat{f}$ trained on the NPLL dataset and the generalization error  $ R( f ^ { * } ) $ of the optimal model  $ f ^ { * } $  diminishes, i.e., **the generalization performance gap between $\hat{f}$  and $f^{*}$ decreases**. The term "converge" might introduce ambiguity in the context. Therefore, we revised the original sentence as "As $n \to \infty$, $\epsilon \to 0$ and $\alpha \to 1$, Theorem 1 shows that the generalized error bound will be reduced, and the empirical risk minimizer $\hat{f}$ will get closer to the true risk minimizer $f^{*}$." and updated the manuscript accordingly.
>
>
>
> **Q2(a):**
>
> ECK is calculated based on the difference (cross-entropy) between CLS-based pseudo-labels ($\boldsymbol{\widetilde{q} _ i}$) and KNN-based pseudo-labels ($\boldsymbol{q _ i}$). For normal samples, since the ground-truth label locates in the CLS, as the model is training, both the KNN-based pseudo-labels and CLS-based pseudo-labels will increasingly approximate the ground truth. Therefore, according to the properties of cross-entropy, a smaller ECK value will be produced. For noisy samples, because the original CLS does not include the ground truth label, while KNN-based pseudo-labels may contain the ground-truth label if the model is well-trained on reconstructed CLS. Consequently, the discrepancy between KNN-based pseudo-labels and CLS-based pseudo-labels (based on the original CLS) is large, resulting in a larger ECK value.
>
> Figure 3 of the paper illustrates the ECK distributions for normal and noisy samples during model training, showing that noisy samples exhibit larger ECK values compared to normal samples. Similar results are observed on another dataset, as presented in Figure 6. Moreover, comparison results in Appendix A.6 demonstrate that the proposed ECK metric outperforms existing NPLL methods in distinguishing noisy samples from normal ones, providing superior separation accuracy.
>
>
>
> **Q2(b):**
>
> We have incorporated a hyper-parameters sensitivity analysis including warm-up epoch into the Appendix A.8. From the experimental results, we observe that the warm-up epoch ($e _ w$) plays a critical role in training stability. Insufficient pretraining with a small $e _ w$ can result in poor feature discrimination, which hinders the generation of reliable pseudo-labels. Conversely, an excessively large $e _ w$ may cause overfitting to noisy samples, reducing the model's ability to correct errors during later training stages. In addition, Table below compares the impact of $e _ w$ under different noise levels on the CIFAR-100 dataset. The results show that as the noise rate increases, the model becomes more sensitive to the choice of $e _ w$, indicating the necessity of careful tuning.
>
> | Dataset\Warm-up epoch                | 100    | 150    | 200    | 250    |
> | ------------------------------------ | ------ | ------ | ------ | ------ |
> | CIFAR100 ( $\eta$=0.05,$\gamma$=0.3) | 79.97% | 80.19% | 80.11% | 79.86% |
> | CIFAR100 ( $\eta$=0.05,$\gamma$=0.4) | 77.77% | 80.13% | 79.43% | 79.41% |
>
> Higher noise rates exacerbate the challenges of the NPLL problem, as evidenced by a significant performance decline across all methods on Table 1 of the manuscript. However, at higher noise levels, the performance gaps between our method and other methods become larger. For example, on CIFAR100 with $\eta=0.1$, our method improves the best compared method by **4.71%**, **8.15%** and **11.57%** on three different noise levels $\gamma=0.2,0.3$ and $0.4$ respectively. The reason is that the proposed sample separation approach can correctly identity the noisy samples, and accordingly relieve the challenges of higher noise rates significantly.

---

> > ### Comment · Reviewer_B6bW · 2024-11-25
> >
> > Thank the authors for their reply. I will maintain the score.
> >
> > Here are a few suggestions for improvement:
> > * Typo in Line 127: "represent" should be corrected to "represents."
> > * Clarification in Line 127: It seems you intended to write "$\Re(\mathcal{H}_y)$" instead of "$\Re(\mathcal{H})$."
> > * Highlighting Revisions: I suggest using a different color to highlight the changes in the revised manuscript. This would make it easier for reviewers to locate the revisions. Once the paper is accepted, you can switch to a unified color for the camera-ready version.
> > * Consider hyper-parameters sensitivity analysis under higher noise level.
> >
> > I trust that the authors will incorporate these suggestions effectively.

---

> > > ### Author Response · Authors · 2024-11-26
> > >
> > > (1-2)
> > >
> > > We appreciate your observation regarding this typo. It has been corrected in the latest version of the paper, and we will conduct a thorough review to avoid the typos in the manuscript.
> > >
> > > (3)
> > >
> > > Thank you for your suggestion regarding highlighting the revisions. In the revised manuscript, we have used **green** to highlight all the changes as per your recommendation. This will help the reviewers easily locate the revisions. Once the paper is accepted, we will ensure that the final camera-ready version uses a unified color format.
> > >
> > > (4)
> > >
> > > Based on your suggestion, we conducted additional parameter sensitivity analysis experiments under higher noise levels on CIFAR100 ( $\eta$=0.05,$\gamma$=0.4), with the results presented in Figure 9 of Appendix A.8. Figures 8 (b, d, f, h) present the results of the parameter sensitivity analysis for our method on the relatively low-noise dataset CIFAR100 ($\eta$=0.05, $\gamma$=0.3). A comparison between Figures 8 and 9 reveals that the hyperparameters $\beta$ and $K$ demonstrate robustness against noise. The model achieves optimal performance when $\beta$ is set between 1.5 and 2 and $K$ is set between 5 and 7 across different noise levels. The hyperparameter $e_w$, representing the warm-up epoch, is relatively susceptible to noise. Specifically, a small $e_w$ can result in poor class discrimination, whereas an excessively large $e_w$ may lead to overfitting on noisy samples. Nonetheless, setting $e_w$ within the range of 100 to 200 consistently yields satisfactory performance across different datasets and noise levels. The choice of $\lambda$ is influenced by the noise rate in the dataset, as $\lambda$ regulates the percentile change rate between normal and noisy samples during the separation process. Therefore, we recommend increasing $\lambda$ under high-noise conditions and reducing it under low-noise scenarios.

---

> ### Author Response · Authors · 2024-11-25
>
> Dear reviewer B6bW,
>
> Thanks again for your time and efforts in reviewing this paper and the valuable comments on improving its quality. As the reviewer-author discussion deadline approaches, please take a few minutes to read the rebuttal. If you have further concerns, we are happy to provide more explanations. Thanks.
>
> Regards from the authors.

---

### Official Review · Reviewer_xLDw · 2024-11-04

**Soundness:** 3
**Presentation:** 3
**Contribution:** 3
**Rating:** 6
**Confidence:** 4

**Summary:**

This paper studies the noisy partial label learning (NPLL) problem, where the correct label is not always in the candidate label set (CLS). This setting is more practice in real world. To address this problem, firstly, the authors theoretically prove the generalization error bound of the classifier constructed under NPLL paradigm and reveal that the error bound depends on the noise rate and the average length of the candidate label set. Secondly, the authors propose a method to reduce the noise rate and reconstruct shorter candidate label sets. Finally, experiments on benchmark datasets are conducted to confirm the efficacy of the proposed method in addressing NPLL.

**Strengths:**

1.	This paper provides a theoretical generalization analysis of the NPLL, highlighting the noise rate and the length of the candidate label sets are two key factors, which is insightful.
2.	The proposed method is grounded in theoretical findings, making it reasonable and intuitive.
3.	Extensive experiments have been conducted and the reported results verify the effectiveness of the proposed method.

**Weaknesses:**

1.	Since the whole label space is very large, then the size of non-CLS may also be very large, which will bring challenges to find the ground truth label.
2.	Some parts of this paper is unclear, see the questions below for details.
3.	Sensitivity analyses of the hyperparameters are absent.

**Questions:**

1.	According to Eq.(2), do you need to know the CLS of the normal samples in prior?
2.	Since the computation of Eq.(3) relies on the KNN-based pseudo-label, how to decide the parameter of K when applying KNN?
3.	In lines 192-195, the authors claim that “when the ground-truth label of a sample locates at the non-candidate labels, there will be a significant discrepancy between the CLS-based pseudo-label and KNN-based pseudo-label, leading to a larger value of ECK”, could you provide some experimental results to support this?
4.	In lines 272-274, the authors claim that “add the label with the highest pseudo-label probability from the non-CLS to the CLS, and remove the label with the lowest pseudo-label probability from the CLS”, this operation may destroy the diversity of CLS. Is it beneficial to the performance? And please explain Eq.(10).
5.	Please give the detailed derivation of second inequality of Eq.(18).
6.	In lines 648-654, there are two repeated references.

---

> ### Author Response · Authors · 2024-11-21
>
> **W1:**
>
> The issue you mentioned is a common challenge for the noisy partial label learning (NPLL). But our proposed method is designed to relieve this challenge. Specifically, we present a candidate label set (CLS) reconstruction approach aimed at generating **shorter** and more precise CLS. This approach facilitates the model's ability to identify ground-truth labels, thereby improving performance. As depicted in Fig. 4(b) of the paper, our method empirically reduces the size of the reconstructed CLS consistently over the training iterations (represented by the green curve). Accordingly, on the dataset with a larger label space, our method achieves a substantially greater performance improvement compared to other baseline methods  e.g., on CIFAR100 ($\eta=0.1$), our method improves the best compared method by **4.71%**, **8.15%** and **11.57%** on three different noise levels $\gamma=0.2,0.3$ and $0.4$ respectively, while on CIFAR10 ($\eta=0.5$), our method performs **2.48%**, **3.83%** and **7.04%** better than the best compared method (Table 1 of the paper).
>
> **W3:**
>
> We appreciate your valuable comments. Following your suggestion, we  have incorporated a hyper-parameter sensitivity analysis into the appendix (A.8) and updated the submitted paper accordingly. Specifically, we have performed sensitivity analysis on four hyper-parameters: the warm-up epoch ($e_w$), the rate of percentile change between $r_l$ and $r_u$ ($\lambda$), the trade-off of two objectives on CLS Reconstruction ($\beta$), and the KNN parameter ($K$).  Results show that $\beta$ and $K$ are stable, with optimal values at $\beta = 1.5/2.0$ and $K = 7/5$, achieving peak accuracies of approximately 95.5%/80.4% for CIFAR-10/100. $\lambda$ performs best at $0.4$ but is relatively sensitive to noise levels in the dataset, requiring careful adjustment. $e_w$ is crucial for training stability, as insufficient warm-up leads to poor pseudo-labeling, while excessive warm-up causes overfitting to noisy labels in the CLS; the optimal value is $150$. Overall, $\beta$ and $K$ are robust, whereas $\lambda$ and $e_w$ need carefully tuning. The detailed analyses and the experimental reulsts can be found in the Appendix A.8 of the paper.
>
>
>
> **Q1:**
>
> In PLL, each sample has a CLS. And in NPLL, each sample remains associated with a CLS, as in PLL. So the CLS of all samples, including both normal samples and noisy ones are accessible. However, different with PLL, some samples' ground-truth labels locate outside the CLS in NPLL, and the these samples (denoted as noisy samples) are mixed with normal ones. The introduction of CLS-based pseudo label computation in Eq. (2) aims to facilitate the distinction between normal samples and noisy samples in subsequent processing stages. Therefore, in Eq. (2), CLS-based pseudo labels are computed for all samples including both the normal ones and noisy ones. In the following subsections, we reconstruct shorter CLS for all samples to replace the initial ones to get the better performance.
>
>
>
> **Q2:**
>
> Our empirical observations indicate that our method achieves consistently high performance with little variation across this interval when the parameter $K$ is set within the range of 5 to 10 on all datasets. Therefore, for the sake of simplicity and consistency, we fixed $K=5$ in all experiments unless the hyper-parameter sensitivity analysis experiments in Section Appendix A.8.
>
>
>
> **Q3:**
>
> Figure 3 of the paper presents the ECK distribution for normal and noisy samples during model training on CIFAR10. The results show that noisy samples (i.e., those with ground-truth labels outside the candidate label set) exhibit larger ECK values compared to normal samples. Furthermore, as training progresses, the gap between the ECK distributions of these two groups of samples becomes more pronounced. This observation supports our claim that the proposed ECK metric effectively distinguishes noisy samples from normal ones. Similar findings are observed on another dataset (CIFAR100), as shown in Figure 6. Additionally, we performed comparisons of separation accuracy with existing NPLL methods in Appendix A.6, demonstrating that the proposed ECK metric is more effective in distinguishing noisy samples from normal ones compared with other NPLL approaches.

---

> ### Author Response · Authors · 2024-11-21
>
> **Q4:**
>
> Yes, this operation will be beneficial to the performance. The detailed explanations and ablation experiments are as follows.
>
> Our dual-threshold method categorizes samples into reliable samples (including reliable normal samples and reliable noisy samples) and uncertain samples. For reliable samples, CLS reconstruction is performed as described in Eqs. (6)–(9). For uncertain samples, their type as either normal sample or noisy sample cannot be well determined. Directly training on those uncertain samples with their original CLS would risk overfitting to label noise, leading to performance degradation. Conversely, discarding uncertain samples entirely would reduce the training sample size, which also negatively impacts performance.
>
> To address the above dilemma, we propose Eq. (10), which tries to refine the CLS of the uncertain samples by swapping the label with the lowest confidence in the CLS and the label with the highest confidence in the non-CLS. This correction mitigates hidden noise in the samples to some extent without increasing the CLS length. It is worth noting that the number of uncertain samples is limited and decreases over iterations (see Figure 3 of the paper), so their impact on the model is also relatively limited. As a summary, the correction of noise is primarily achieved through CLS reconstruction by Eqs. (6)-(9), while the main role of Eq. (10) is to prevent the model from overfitting to the noise within these uncertain samples.
>
> We validated the effectiveness of Eq. (10) through experiments on CIFAR100 ($\eta = 0.05, \gamma = 0.3$) and CIFAR100 ($\eta = 0.05, \gamma = 0.4$), with the results presented in the table below. Here, *Ours* represents our proposed method, *Ours Original* uses the original CLS for uncertain samples during training, and *Ours Discard* discards uncertain samples entirely. The results show that our method consistently achieves the highest performance. Notably, under higher noise rates, the gap between *Ours* and *Ours Original* widens, further highlighting the effectiveness of our approach.
>
> | Method\Dataset  | CIFAR100 ( $\eta$=0.05,$\gamma$=0.3) | CIFAR100 ( $\eta$=0.05,$\gamma$=0.4) |
> | --------------- | ------------------------------------ | ------------------------------------ |
> | *Ours*          | **79.88%**                           | **79.29%**                           |
> | *Ours Original* | 79.56%                               | 78.77%                               |
> | *Ours Discard*  | 78.68%                               | 78.43%                               |
>
>
>
> **Q5:**
>
> The first inequality of Eq. (18) is
> $$
> \widehat{R}^{\prime}(f)  \geq \widehat{R}(f) - \frac{1}{n} \sum _ {i=1}^{n} \mathbb{I}(\boldsymbol{y} _ {i} \in Y _ {i}) \frac{\left| Y _ {i} \right| - 1}{\left| Y _ {i} \right|} \mathcal{L}(f(\boldsymbol{x} _ {i}),\boldsymbol{y} _ {i}) - \frac{1}{n} \sum _ {i=1}^{n} \mathbb{I}(\boldsymbol{y} _ {i} \notin Y _ {i}) \mathcal{L}(f(\boldsymbol{x} _ {i}),\boldsymbol{y} _ {i})
> $$
>
> Since $\mathcal{L}(f(\boldsymbol{x}), y)$ is upper-bounded by $M$ ($\mathcal{L}(f(\boldsymbol{x}), y) \le M$), we can deduce
>
> $$
> \widehat{R}^{\prime}(f)  \geq \widehat{R}(f) - \frac{1}{n} \sum _ {i=1}^{n} \mathbb{I}(\boldsymbol{y} _ {i} \in Y _ {i}) \frac{\left| Y _ {i} \right| - 1}{\left| Y _ {i} \right|} M - \frac{1}{n} \sum _ {i=1}^{n} \mathbb{I}(\boldsymbol{y} _ {i} \notin Y _ {i}) M.
> $$
>
> Define $m = \sum _ {i=1}^{n} \mathbb{I}(\boldsymbol{y} _ i \in Y _ i)$, which implies the noise rate $\epsilon = \frac{n - m}{n}$. Define the function $f(|Y|) = \frac{|Y| - 1}{|Y|}$, which is concave. By Jensen's inequality and the definition of $\alpha$ ($\alpha=\frac{1}{m} \sum _ {j=1}^{m} |Y _ j|$), we have
>
> $$
> \frac{1}{n} \sum _ {i=1}^{n} \mathbb{I}(\boldsymbol{y} _ {i} \in Y _ {i}) f(|Y _ i|)
> $$
>
> $$
> = \frac{m}{n} \cdot \frac{1}{m} \sum _ {j=1}^{m} f(|Y _ j|)
> $$
>
> $$
> \leq \frac{m}{n} f\left(\frac{1}{m} \sum _ {j=1}^{m} |Y _ j| \right)
> $$
>
> $$
> = (1 - \epsilon) f(\alpha).
> $$
>
> Combing the above two inequalities, we have
>
> $$
> \widehat{R}^{\prime}(f) \geq \widehat{R}(f) - (1 - \epsilon) \frac{\alpha - 1}{\alpha} M - \frac{1}{n} \sum _ {i=1}^{n} \mathbb{I}(\boldsymbol{y} _ {i} \notin Y _ {i}) \mathcal{L}(f(\boldsymbol{x} _ {i}),\boldsymbol{y} _ {i}).
> $$
>
> At this point, the second line of Eq. (18) is complete.
>
>
>
> **Q6:**
>
> Thank you very much for pointing out the typo, which have been corrected in the revised version.

---

> ### Author Response · Authors · 2024-11-25
>
> Dear reviewer xLDw,
>
> Thanks again for your time and efforts in reviewing this paper and the valuable comments on improving its quality. As the reviewer-author discussion deadline approaches, please take a few minutes to read the rebuttal. If you have further concerns, we are happy to provide more explanations. Thanks.
>
> Regards from the authors.

---

> > ### Comment · Reviewer_xLDw · 2024-11-25
> >
> > Thanks for your detailed response, the derivation of Eq.(19) is still confusing, please detail the derivation of it.

---

> > > ### Author Response · Authors · 2024-11-26
> > >
> > > The first line of Eq. (19) is
> > > $$
> > > \widehat{R}^{\prime}(f) \leq \widehat{R}(f) + \frac{1}{n} \sum _ {i=1}^{n} \mathbb{I}(\boldsymbol{y} _ {i} \in Y _ {i}) \sum _ {c \in Y _ {i}, c \neq \boldsymbol{y} _ {i}} \frac{1}{\left| Y _ {i} \right|} \mathcal{L}(f(\boldsymbol{x} _ {i}),c) + \frac{1}{n} \sum _ {i=1}^{n} \mathbb{I}(\boldsymbol{y} _ {i} \notin Y _ {i}) \sum _ {c \in Y _ {i}} \frac{1}{\left| Y _ {i} \right|} \mathcal{L}(f(\boldsymbol{x} _ {i}),c)
> > > $$
> > >
> > > Since $\mathcal{L}(f(\boldsymbol{x}), y)$ is upper-bounded by $M$ ($\mathcal{L}(f(\boldsymbol{x}), y) \le M$). Define $m = \sum _ {i=1}^{n} \mathbb{I}(\boldsymbol{y} _ i \in Y _ i)$, which implies the noise rate $\epsilon = \frac{n - m}{n}$. we can deduce
> > >
> > > $$
> > > \widehat{R}^{\prime}(f) \leq \widehat{R}(f) + \frac{1}{n} \sum _ {i=1}^{n} \mathbb{I}(\boldsymbol{y} _ {i} \in Y _ {i}) \sum _ {c \in Y _ {i}, c \neq \boldsymbol{y} _ {i}} \frac{1}{\left| Y _ {i} \right|} M + \frac{1}{n} \sum _ {i=1}^{n} \mathbb{I}(\boldsymbol{y} _ {i} \notin Y _ {i}) \sum _ {c \in Y _ {i}} \frac{1}{\left| Y _ {i} \right|} M
> > > $$
> > >
> > > $$
> > > = \widehat{R}(f) + \frac{1}{n} \sum _ {i=1}^{n} \mathbb{I}(\boldsymbol{y} _ {i} \in Y _ {i})  \frac{| Y _ {i}| - 1}{\left| Y _ {i} \right|} M + \frac{1}{n} \sum _ {i=1}^{n} \mathbb{I}(\boldsymbol{y} _ {i} \notin Y _ {i}) M
> > > $$
> > >
> > > $$
> > > = \widehat{R}(f) + \frac{m}{n} \cdot \frac{1}{m} \sum _ {j=1}^{m}  \frac{| Y _ {j}| - 1}{\left| Y _ {j} \right|} M + \frac{n-m}{n} M
> > > $$
> > >
> > > $$
> > > = \widehat{R}(f) + (1 - \epsilon) \frac{1}{m}\sum _ {j=1}^{m}  \frac{| Y _ {i}| - 1}{\left| Y _ {i} \right|} M + \epsilon M.
> > > $$
> > >
> > >
> > >
> > > Define the function $f(|Y|) = \frac{|Y| - 1}{|Y|}$, which is concave. By Jensen's inequality and the definition of $\alpha$ ($\alpha=\frac{1}{m} \sum _ {j=1}^{m} |Y _ j|$), we have
> > > $$
> > > \frac{1}{m} \sum _ {j=1}^{m} f(|Y _ j|) \leq  f\left(\frac{1}{m} \sum _ {j=1}^{m} |Y _ j| \right) = f(\alpha).
> > > $$
> > >
> > > Combing the above two inequalities, we have
> > >
> > > $$
> > > \widehat{R}^{\prime}(f) \leq \widehat{R}(f) + (1-\epsilon) \frac{\alpha-1}{\alpha} M + \epsilon M
> > > $$
> > >
> > > $$
> > > = \widehat{R}(f) + (1 - \frac{1 - \epsilon}{\alpha}) M
> > > $$
> > >
> > >
> > >
> > > At this point, the Eq. (19) is complete.
> > >
> > > Thank you for your valuable feedback. In response, we have revised and updated the proof details of Eq. (18) and Eq. (19) in the latest version of the manuscript to enhance clarity.

---

> > > > ### Comment · Reviewer_xLDw · 2024-11-26
> > > >
> > > > Thank you for your response. The derivations of Eq. (18) and Eq. (19) are now much clearer, and I have no more questions. I will keep my positive score.

---

### Author Response · Authors · 2024-11-23

We sincerely thank all reviewers for their time and constructive comments. We are delighted that the reviewers recognize the value of our work and provided positive feedback regarding its contributions:

- **Theoretical contributions:** 1. "This paper provides a **theoretical generalization analysis of the NPLL**, highlighting ..., which is **insightful**." (xLDw) 2. "**provide the generalization error bound** of the classifier constructed under NPLL" (1LBQ, RE2k)  3. "**a significant theoretical advancement.**" (Xu7T)
- **Novel methodological design:** 1. "The proposed method is **grounded in theoretical findings**, making it **reasonable and intuitive**" (xLDw, B6bW) 2. "**their method can effectively** reduce ..." (1LBQ) 3. " **innovative approach to solving NPLL.**" (Xu7T)
- **Comprehensive experiments:** 1. "Extensive experiments ... verify the **effectiveness** of the proposed method." (xLDw) 2. "The performance of the proposed method is **promising** in the experiments." (B6bW) 3. "**significantly outperforms** the current state-of-the-art (SOTA) methods." (1LBQ, RE2k) 4. "The experiments are generally **comprehensive**." (Xu7T)
- **Clear and well-structured writing:** 1. "This paper is **clearly written and easy to understand**." (B6bW, RE2k) 2. "The article's structure is **well-organized** and easy to comprehend." (Xu7T)

We are deeply grateful for the reviewers’ encouraging feedback and insightful suggestions, which motivate us to further refine our work. We remain open to any further questions or discussions related to this research. Thank you again for your valuable time and input!

---

### Meta-Review · Area_Chair_29Hc · 2024-12-24

**Metareview:**

The paper addresses the problem of noisy partial label learning (NPLL) and received five reviews with ratings of 6, 6, 6, 5, and 6. It presents a novel theoretical generalization analysis for NPLL, laying the foundation for a simple yet effective NPLL learning framework. The paper is well-written, and the proposed approach is both theoretically sound and practically effective. Under this framework, training samples are progressively separated to reduce the noise rate while reconstructing shorter CLS. The proposed framework is versatile, as it can be applied to many existing methods, and demonstrates strong empirical performance, consistently outperforming state-of-the-art approaches. This is a solid contribution on NPLL. During the rebuttal phase, the authors effectively addressed the reviewers' concerns. Based on the overall positive feedback, this paper is recommended for acceptance. The authors are expected to incorporate the reviewers' revision suggestions in the final version of the paper.

**Additional Comments On Reviewer Discussion:**

Most reviewers participated in the discussions with the authors during the rebuttal and their concerns are addressed.

---

### Decision · Program_Chairs · 2025-01-22

Accept (Poster)